# Self-Assembly of Amphiphilic Compounds as a Versatile Tool for Construction of Nanoscale Drug Carriers

**DOI:** 10.3390/ijms21186961

**Published:** 2020-09-22

**Authors:** Ruslan Kashapov, Gulnara Gaynanova, Dinar Gabdrakhmanov, Denis Kuznetsov, Rais Pavlov, Konstantin Petrov, Lucia Zakharova, Oleg Sinyashin

**Affiliations:** A.E. Arbuzov Institute of Organic and Physical Chemistry, FRC Kazan Scientific Center of RAS, Arbuzov street 8, Kazan 420088, Russia; ggulnara@bk.ru (G.G.); nemezc1988@yandex.ru (D.G.); kuznetsov_denis91@mail.ru (D.K.); rais.pavlov@iopc.ru (R.P.); kpetrov2005@mail.ru (K.P.); lucia@iopc.ru (L.Z.); oleg@iopc.ru (O.S.)

**Keywords:** amphiphile, cationic surfactants, drug delivery, liposome, endosomal escape, macrocycle, polymer, mucoadhesion

## Abstract

This review focuses on synthetic and natural amphiphilic systems prepared from straight-chain and macrocyclic compounds capable of self-assembly with the formation of nanoscale aggregates of different morphology and their application as drug carriers. Since numerous biological species (lipid membrane, bacterial cell wall, mucous membrane, corneal epithelium, biopolymers, e.g., proteins, nucleic acids) bear negatively charged fragments, much attention is paid to cationic carriers providing high affinity for encapsulated drugs to targeted cells. First part of the review is devoted to self-assembling and functional properties of surfactant systems, with special attention focusing on cationic amphiphiles, including those bearing natural or cleavable fragments. Further, lipid formulations, especially liposomes, are discussed in terms of their fabrication and application for intracellular drug delivery. This section highlights several features of these carriers, including noncovalent modification of lipid formulations by cationic surfactants, pH-responsive properties, endosomal escape, etc. Third part of the review deals with nanocarriers based on macrocyclic compounds, with such important characteristics as mucoadhesive properties emphasized. In this section, different combinations of cyclodextrin platform conjugated with polymers is considered as drug delivery systems with synergetic effect that improves solubility, targeting and biocompatibility of formulations.

## 1. Self-Assembly of Amphiphilic Compounds

Self-assembling systems based on amphiphilic compounds find wide application in different fundamental and practical areas due to their unique ability to form nanoscale assemblies with gradients of polarity, viscosity, electric charge and other characteristics [1,2,3]. In an aqueous medium, such nanosized aggregates possess a nonpolar interior capable of entrapping guest molecules, thereby dramatically changing their properties [2,3,4]. This phenomenon is responsible for the wide range of applications of amphiphilic compounds in cosmetics, food industry, pharmacy, drug and gene delivery, etc. [5,6,7,8]. Meanwhile, for these reasons they are strongly required to meet the green chemistry criteria. Therefore, the design of environmentally friendly amphiphilic compound (surfactants, macrocycles, polymers) is a challenging task from the viewpoint of the development of supramolecular multifunctional systems with tunable characteristics.

### 1.1. Cationic Surfactants Bearing Cleavable Fragments

Among amphiphilic compounds, special attention is received by cationic surfactants, which is related to both their fundamental and practical relevance. Researchers have succeeded in designing and synthesizing a variety of homologous series of cationic surfactants differing in the structure of head groups, which allows for determining the role of the polar fragment in the physicochemical properties and functional activity of supramolecular systems based on cationic surfactants in terms of structure–properties–activity relationships. Numerous fundamental approaches and practical applications involving cationic surfactants have recently been demonstrated in the fields of biochemistry, nanomedicine, pharmacy, catalysis, corrosion protection, oil recovery, food, cosmetics, etc. [2]. These beneficial results were achieved despite the well-known toxicity of cationic surfactants, since much effort has been undertaken to overcome this limitation. Therefore, the design of novel cationic surfactants addressing this problem is of importance. To this end: (i) cationic surfactants with specific structures, including cleavable, gemini, and biocompatible surfactants have been designed; (ii) mixed compositions with less toxic nonionic surfactants are used; (iii) modifications with hydrotropic agents are carried out. Some of these studies devoted to synthesis and use of novel amphiphilic compounds answering biotechnological criteria are discussed below, with emphasis on cationic surfactants.

Recently, much attention has been received by the so-called cleavable surfactants bearing ester, amide, disulfide or other degradable fragments [9]. While researchers initially focused on ester quats bearing quaternized nitrogens in the polar group [10], more diverse amphiphilic structures have been designed, including those containing a carbamate moiety [11], gemini analogs [12], or amphiphilic matrices bearing two cleavable groups [13]. The structural characteristics of ester quats are of importance in their surface activity, with synergetic hydrolysis-driven effects being observed for derivatives in which the carbonyl groups are bridged by an oxygen atom with a quarternized nitrogen atom [10]. Series of gemini surfactants bearing ester and amide groups were synthesized and systematically studied in [14], with exploration of their degradability, surface activity, foaming and antimicrobial properties.

A series of publications [15,16,17,18] focused on carbamate-bearing surfactants in single and mixed systems, including those containing additionally an imidazolium moiety.

For these carbamate surfactants, critical micelle concentration (CMC) values were shown to be lower compared to conventional cationic surfactants with trimethylammonium (TMA) head groups and to decrease with an increase in the number of carbon atoms (n) in the alkyl tail as follows: lg (CMC) = 0.563–0.257 × n [15]. A series of carbamate surfactants (Figure 1a) was tested for their solubilization capacity toward hydrophobic probes and antimicrobial activity, which confirm them as biodegradable micellar nanocontainers with improved functional efficacy. The hexadecyl derivative was further investigated in composition with nonionic surfactants, with negative deviation revealed from ideal mixture behavior [13,16]. These mixed systems demonstrated effective solubilization properties toward the anti-inflammatory drug meloxicam, with a balance achieved between the low toxicity characteristics due to the presence of nonionic surfactants and the solubilization activity maintained by cationic surfactants. In [17] novel formulations for the herbicide Lontrel^®^ were developed with the use of a series of cationic surfactants. Carbamate-bearing surfactants were found to show the highest activity, with a 3-fold increase in the Lontrel^®^ concentration observed in the plants tested and improved wetting ability compared to an unformulated preparation. These results could be further enhanced by increasing the temperature. The biomedical functionality of carbamate-bearing surfactants could be significantly modified by the introduction of an imidazolium fragment [18]. For a homologous series of imidazolium-containing amphiphiles bearing carbamate groups (Figure 1b), antimicrobial properties within the concentration range characterized by low hemolytic activity are documented. In addition, membranotropic ability is revealed, which is more pronounced in the case of lower homologs. A valuable property of cationic surfactants is their affinity to negatively charged biological species, which is responsible, for example, for their application as nonviral vectors. The complexation activity of imidazolium amphiphiles with carbamate fragments toward DNA decamer and bovine serum albumin (BSA) macromolecules was verified by a variety of techniques and treated in terms of their electrostatic interaction and hydrogen bonding ability [18].

### 1.2. Amphiphilic Compounds with Natural Fragment

Among the wide spectrum of recently synthesized amphiphilic compounds a special place is occupied by amphiphiles bearing natural fragments in their structures (Figure 2). The main idea behind the design of this type of amphiphilic compounds is based on principles of biomimetics, that allow constructing environmentally and physiologically friendly supramolecular systems of nanoscale dimensions for a number of practical applications in industry and biomedicine.

Since the term “natural fragment” is quite common and could include a lot of different structural fragments, it is practically impossible to overview all possible combinations in the framework of this review. Therefore, only selected amphiphiles containing natural moieties are discussed below, namely: (i) amphiphilic compounds bearing amino acid fragments; (ii) hydrophobized derivatives of saccharides of different types; (iii) salts of bile acids and their derivatives and (iv) nucleolipids. In line with the biomimetic approach, the introduction of these residues is assumed to improve the biocompatibility and reduce the toxic characteristics of amphiphilic molecules, to contribute to complementary interactions when using in vivo assays, to allow for realizing various aggregate morphological structures (Figure 3), etc.

#### 1.2.1. Amphiphilic Compounds Bearing Amino Acid Fragments

One of the most popular routes for modification of amphiphilic compounds and making systems on their basis more biocompatible and bioavailable is the introduction of amino acid fragments into their chemical structures [19]. These synthetic amino acid-based surfactants are an alternative to ordinary antimicrobial compounds, because they can be obtained from renewable raw materials.

Amino acid-based amphiphiles are promising in obtaining drugs based on surfactant–protein complexes [19] and can be used in oil recovery [20], as nonviral vectors in gene therapy [21], in cosmetics [22], etc. The main requirements for the resulting amphiphiles are multifunctionality, low toxicity and biodegradability [23].

It is well known that in accordance with one of the conventional classifications of amino acids from the viewpoint of their charge characteristics provided by the nature of their substituents, they could be divided into four big groups [24]: (1) nonpolar amino acids; (2) polar amino acids; (3) polar acidic amino acids (negatively charged); (4) polar basic amino acids (positively charged). Since the charge of an amphiphilic compound determines its properties to a great extent, the corresponding derivatives of amino acids should be discussed from this perspective.

The main attention of researchers is focused on the application of amphiphilic compounds bearing nonpolar amino acids. Although there are several examples of studies dealing with the synthesis of this kind of compounds [25], the majority of investigations are dedicated to discussion of their physicochemical properties in solutions and potential applications in biomedicine. For example, hydrophobized phenylalanine and valine derivatives demonstrated high cytotoxicity against the MCF-7 and HEK cell lines [26], pH-dependent self-assembly behavior [27] and could be used for construction of worm-like aggregates for ibuprofen delivery [28]. As a separate line of investigation, the interaction of these amphiphiles with biological species should be mentioned: glycine and sarcosinate derivatives are approved as effective agents for complexation with carrageenans of various types [29] and myoglobin [30]. Besides, compositions based on amphiphiles with amino acid fragments exhibit controlled morphological [31] and rheological [32] behavior, valuable in oil recovery processes, for antimicrobial activity and suitable biodegradability [33]. Alongside with these reports, there are systematic investigations of the role of amino acid structures in various properties: it has been shown that variation of the structure of amino acid (proline, phenylalanine, isoleucine) could be used as a tool for regulating of the stability of emulsions based on them and antibacterial activity against *Staphylococcus aureus* and *Bacillus subtilis* strains [34].

Amphiphiles bearing polar amino acids and polar acidic amino acids (negatively charged) have attracted less attention. Polar amino acid derivatives are represented by serine- and cysteine-containing amphiphiles exhibiting a more than one order of magnitude lower hemolytic activity in comparison with conventionally used sodium dodecyl sulphate [35]. They demonstrated biomedical potential, which can be exemplified by their study as nonviral vectors for delivery of DNA into HeLa cancer cells with high gene transfection efficiency (up to 50%) [36]. Their application in the synthesis of catalytic active modified silver nanoparticles for *p*-nitrophenol reduction was documented in [37]. The synthesis of a new cationic double-tailed cysteine-based surfactant is given in [38]. For this amphiphile, the aggregation characteristics and the ability to bind to BSA were evaluated. It was shown that there are various intermolecular interactions between the components depending on the surfactant concentration. 

Investigations of amino acid derivatives belonging to the negatively charged amino acid class are exemplified herein by arginine-containing amphiphilic compounds documented as additives for modification of physicochemical properties of oil [39], potential antimicrobial compounds active against *Micrococcus luteus*, *Bacillus subtilis*, *Staphylococcus aureus* strains [40] and agents capable to bind with heparin [41], which activity could be regulated by selecting appropriate alkyl tail length for the amphiphile. Compositions based on amphiphilic lysine derivatives were documented as drug delivery vehicles containing encapsulated 5-fluorouracil [42] for cancer treatment as well as the component for design of coating against protein adsorption on nanoparticles [43]. In [44] promising stimuli-responsive double-tailed lysine-based surfactants were obtained. 

Below the Krafft point, the amphiphiles self-organized into tubular structures of various morphologies forming hydrogels at low surfactant concentration; above the Krafft point, there was a transition from tubular structures to micelles or vesicles, depending on the structure of the surfactant, and this transition was thermo-reversible in the physiological temperature range. The synthesis of histidine-based surfactants with high antimicrobial activity against Gram-positive and Gram-negative bacteria is given in [45], with some of them displaying no destructive effect on red blood cells. The work [46] is devoted to the synthesis of arginine-based surfactants and the study of their hemolytic activity. It was found that this amphiphile protected human red blood cells from hypotonic lysis over a wide range of concentrations. Dicationic histidine derivatives were documented as promising agents for antibacterial treatment against new generations of microorganisms like methicillin-resistant *Staphylococcus aureus* [47]. The authors of [48] investigated the surface-active and micelle-forming properties of three anionic dicarboxylic amino acid-based surfactants in combination with the cationic compound cethyltrimethylammonium bromide (CTAB). It was shown that maximum number of the cationic surfactant CTAB molecules on the micellar surface leads to the formation of close-packed micellar structures [49]. The CMC values were significantly lower than predicted, which indicates the existence of associative interactions between the components.

Several examples of amino acid-based amphiphiles and their properties are given in Table 1. Analysis of these values allows one to conclude that the aggregation and antimicrobial properties of this type of amphiphiles could be significantly enhanced by the introduction of additional head groups and hydrophobic tails into the chemical structures of molecules, as well as through covalent linking of acyclic NH-fragments. The latter could be due to the increase of the hydrogen bonding contribution to aggregation processes and destruction of bacterial cell wall due to additional amine fragments.

#### 1.2.2. Sugar-Based Amphiphilic Compounds

Another promising approach to design substances meeting biomedical application criteria is the functionalization of amphiphiles by a saccharide moiety [50], that makes it possible to prepare sugar-based surfactants, such as alkylpolyglycosides, alkylglucamines, etc. Among the obvious advantages of these substances are their biodegradability, low toxicity and low cost. This research direction looks so promising, that researchers have used significant theoretical approaches to predict their aggregation parameters [51]. An overview of the reports dedicated to sugar-containing amphiphiles allows one to deduce that these compounds could be classified by two ways from the viewpoint of the degree of oligomerization of the sugar moiety inserted in the chemical structure of an amphiphilic compound. The first classification takes into account the number of furanose and/or pyranose moieties in the structure and therefore the compounds can be divided into three groups: (1) amphiphiles bearing monosaccharide fragments; (2) amphiphilic compounds containing disaccharide moieties; (3) hydrophobized derivatives of trisaccharides. Another classification is based on the nature of the head group and the number of hydrophobic tails and head groups of amphiphiles, so these derivatives could be divided into: (i) monomeric nonionic surfactants, (ii) nonionic gemini surfactants and (iii) cationic surfactants and other amphiphiles with a sugar fragment. Herein, we consider some general properties for each of the six mentioned groups.

The majority of monosaccharide amphiphilic compounds are represented by glucose derivatives. Interestingly, the selection of functionalization of glucose-containing amphiphilic compounds and using of various approaches to molecule design could become a useful tool for tailoring desired practical properties. For example: (i) significant foamability and high emulsion stability for toluene/water system could be achieved in the case of application of amido- and/or alkoxy derivatives of this hexose [52]; (ii) introduction of a trisiloxane fragment into the glucose structure is responsible for superspreading properties on hydrophobic surfaces like parafilm [53]; (iii) the transition from monomeric amphiphiles to their gemini analogues containing two glucose fragments and additional OH-groups is key for the construction of vehicles for biomedicine purposes capable of effectively encapsulating the drug of a broad spectrum, resveratrol (encapsulation efficiency reaches 90%) [54], and amino acids with demonstrated of chiral specificity toward certain D- or L-isomers [55]; (iv) thermotropic liquid crystalline behavior can be achieved in the case of sulfur-containing glucose derivatives [56]. The utilization of other monosaccharides as building blocks for amphiphilic compounds is also well-known, however these reports are significantly fewer: mannose derivatives are documented as agents for regulating the aggregation state of β-lactoglobulin [57], and a xylopyranose-based amphiphilic compound shows itself as a material for the construction of liquid crystals [58].

Disaccharide derivatives cover amphiphiles mainly containing lactose, maltose and sucrose fragments in their structures. The presence of a lactose moiety in the molecule provides both industrial and biotechnological significance to these compositions. These compounds could be used as efficient micellar catalysts of Ullmann C-S coupling reactions in water (the yield of product could reach 90% by selecting an appropriate alkyl tail length) [59]; highly-effective antibacterial agents against a wide spectrum of microorganisms [60]; coatings interrupting undesired adsorption processes [61]; compounds for condensation with DNA, that could be accurately tuned by selection of the counterion [62]; a basis of a vehicle for ketoprofen delivery [63]. Maltose and sucrose amphiphilic derivatives were recommended as components of microemulsions—potential alternative fuel [64], regulators of wettability of various solid surfaces like quartz and polyethylene [65] and modifiers of adsorption parameters at the air/liquid interface of systems [66]. Herein, crocin, a natural bolaamphiphile with two disaccharide moieties, should be separately mentioned. Recently this carotenoid was used for micellar catalysis of oxidation processes [67]. Besides the abovementioned disaccharides, a sophorose amphiphilic derivative was reported as a building block for the construction of unique supramolecular architectures, namely vesicle-in-vesicle units with diameters up to 200 nm [68].

There are only a couple of recent applications dealing with trisaccharide amphiphilic compounds, nevertheless compositions based on them show themselves to be biocompatible and suitable for biotechnological purposes. For example, raffinose and melezitose amphiphilic derivatives was documented as effective solubilizers for polycyclic aromatic hydrocarbons possessing acceptable hemolytic activity, which could be reduced by shortening the amphiphile alkyl tail [69] as well as agents allowing one to maintain the activity of enzymes during 20 days after its thermal treatment [70]. 

A series of alkyltriazole glycosides (ATGs) with different length of spacer linking the sugar and triazole moieties were synthesized, and their surface-active properties and phase behavior were studied [71]. It was shown that the ATG derivative with the shortest spacer length forms a bicontinuous cubic phase and the position of the triazole bond in ATG doesn’t significantly affect the surface activity.

In [72] the synthesis of lactose-based non-ionic surfactants and their analysis in vitro were accomplished. The cytotoxic activity of these amphiphiles toward two cell lines, colonic epithelium Caco-2 and airway epithelium Calu-3, was studied. It was shown that cell apoptosis occurred at a low concentration of the surfactants.

The effect of three different mannose-based surfactants included in the lipid bilayer on the physicochemical parameters of the system was studied in [73]. It was shown that the mannose derivatives differ in properties from other alkylmonosaccharides due to their structural features. For example, octyl and dodecyl derivatives integrate into hydrophilic regions of the lipid bilayer, while a hexadecyl derivative modifies the internal and external regions of the model membrane.

In [74] the effect of octyl-D-glucopyranoside-based surfactants on the environment was investigated. The study showed that they were completely biodegradable with different utilization efficiencies depending on the functional group. It is important to note that these amphiphiles were toxic towards Gram-negative bacteria cells due to a decrease in zeta potential, thereby increasing the permeability of the cell membrane.

Gemini surfactants containing two hydrophilic head groups and two hydrophobic tails exhibit beneficial physicochemical properties, in particular, they have significantly reduced CMC values and increased surface activity compared to monomeric amphiphiles. In the works of Liu et al. [75], gemini alkyl glucosides were synthesized and studied. In addition to increasing the surface activity, interesting self-assembly features were observed. The gemini dodecyl O-glucoside-based surfactants demonstrated high pass-through capacity for vesicles loaded with (+)-catechin and (–)-epigallocatechin. The incorporation of these drugs into the vesicle made it possible to strengthen the vesicular bilayer in the low concentration range.

In [76] glucono-δ-lactone-based gemini surfactants were studied. For these compounds, a decrease in CMC and more efficient adsorption at the air-water interface were observed compared to monomeric derivatives. Depending on the length of the alkyl tail and concentration, they can form both large aggregates (vesicles), and small aggregates of different sizes (micelles).

Cationic surfactants are mentioned above to be an important class of chemical compounds, which find wide practical application as emulsifiers in cosmetology, bactericidal agents, corrosion inhibitors, antifoams and in medicine. At present, many studies are focused on the increase of biocompatibility and biodegradability of cationic surfactants. To achieve these goals, sugar-based cationic surfactants are obtained. Glucose-based cationic surfactants containing an ester group were obtained in [77], and their surface properties and antimicrobial activity were studied. These amphiphiles are characterized by a decrease in surface tension and CMC in comparison with conventional surfactants, as well as excellent compatibility with sodium dodecyl sulfate.

In [78] reverse micelles formed by sugar-based surfactants with dicarboxylate moieties as a counterion were used to extract BSA. Under optimal conditions, almost complete dissolution of BSA in the micelles of surfactants with a glucosylammonium fragment was observed, while 70–80% of BSA was dissolved in micelles of surfactants with a lactosylammonium head group. The influence of various system parameters on the morphology of aggregates of cationic gemini surfactants containing an isosorbide fragment was studied in [79]. With a change in pH or by heating the system, aggregates were rearranged from micelles to vesicles, and vice versa, and in the presence of salts a transition from vesicles to micelles was observed.

Analysis of existing representatives of sugar-based amphiphiles has demonstrated that key criteria responsible for the best aggregation characteristics (the lowest CMC values) of sugar-based surfactants are the presence of more than one hydrophobic tail in the structure and the transition from monosaccharide to oligosaccharide derivatives. If the first mentioned criterion is very similar to conventional surfactants and has the same nature, the second one could be due to allowing for more intermolecular hydrogen bonds between the OH-groups of additional saccharide moieties. Several examples of amphiphiles with superior self-assembling propensity are listed in Table 2.

#### 1.2.3. Bile Salts and Derivatives of Bile Acids—Natural Amphiphiles and Their Artificially Designed Counterparts

Salts of bile acids are derived from steroid acids contained in the bile of mammals. Their main difference from conventional amphiphiles is the presence of a bulky scaffold, which predetermines their unique properties. For example, bile salts are known as effective transporters of lipids [81], regulators of the rate of crystallization [82] and components of compositions aimed at drug delivery [83]. The most widespread bile salts considered in recent reports are cholate, deoxycholate and taurodeoxycholate. Sodium cholate is known for its capability to act as a transdermal enhancer [84] and additive to mono- and dicationic surfactant solutions significantly modifying their morphological behavior [85] and release profile of cargo encapsulated in self-assembled drug nanovehicles [86]. Sodium deoxycholate was reported as part of biamphiphiles, ionic compounds with both cationic and anionic amphiphilic nature, which aggregate shape could be accurately controlled [87]; micellar catalyst for the synthesis of 1,2,3-triazole derivatives; their characteristics is superior in comparison with those of conventional surfactants CTAB and Triton-X-100 [88]. Sodium taurodeoxyholate was documented as a compound for regulation of solubility of steroid drug in surfactant micellar solutions and biorelevant media [89] and as a component of composition for albendazole delivery for the treatment of lung parasites [90]. 

Further attempts of bile acids functionalization have been presented in recent reports. There are a couple of reports discussing *p*-tertbutylbenzoyl derivatives of cholic acid, which when mixed with a copolymer and in a covalent conjugation with tripeptide demonstrate fabrication of thermoresponsive supramolecular structures [91] and twisted nanoribbons [92] as well as dimeric bile acid derivatives acting as organogelators for the design of hybrid nanomaterials [93].

#### 1.2.4. Nucleolipids (Amphiphilic Derivatives of Nucleobases)

A special place among amphiphiles bearing natural fragments is occupied by the so-called nucleolipids–hydrophobic derivatives of nucleobases, nucleosides and nucleotides. Interest in these compounds is mostly due to the nitrogenous bases (adenine, uracil, thymine, guanine) inserted in their chemical structure and capable of being involved in additional specific interactions. As an example, nucleolipids was known by their capability to form films at air/liquid interfaces [94], coatings for surface hydrophobization [95], liquid crystalline mesophases [96], as well as they have potential for the design of highly-sensitive sensors for nucleobases [97] and inorganic ions [98]. Besides, the specific architecture of nucleolipid molecules and their biocompatibility allows their use for biomedical purposes as neurotracers [99], drugs against glioblastoma [100] and carriers for potential anti-malarial drugs [101]. Systematic studies [102,103,104,105] have focused on charged representatives of this type of amphiphilic compounds (Figure 4), exemplified by a series of cationic nucleolipids bearing a pyrimidine moiety, which allows revealing quantitative structure–activity relationships. For example, it has been established that these nucleolipids have unique aggregation behavior in aqueous solutions [102], solubilization properties toward model hydrophobic compounds [103], marked membranotropic properties [104] and substrate-specific effects on the catalysis of ecotoxicants of various hydrophobicity [105].

A comparative analysis of collected data for nucleolipids of various structure (Table 3) was conducted in order to establish the functional significance in their amphiphilic structures. The main features responsible for increase in aggregation and solubilization properties of pyrimidine-based nucleolipids are the introduction of a hydrophobic tail into the uracil residue, the growth of hydrophobicity of the cyclic head group, as well as transition from acyclic derivatives to macrocyclic ones and insertion of additional nucleobases and hydrophobic tails to the chemical structure of the molecule.

### 1.3. Silicone-Based Surfactants

Silicone-based surfactants are new effective amphiphiles with advanced characteristics. They are widely used in the leather industry [113], agriculture [114], as cosmetic emulsifiers, fabric softeners [115], etc. In addition, cationic silicone-based gemini surfactants are in accordance with the principles of green chemistry due to low toxicity and quick decomposition of the Si-O-Si bond [116]. Importantly, silicone-based amphiphiles reduce surface tension better than hydrocarbon surfactants.

The presence of a tetrasiloxane fragment in the structure of cationic gemini surfactants can potentially improve the biocompatibility and various characteristics of amphiphiles. A series of new cationic gemini surfactants with a tetrasiloxane fragment (Figure 5a) was synthesized in [117]. The formation of various aggregates (vesicles, spherical micelles and rod-like micelles) depending on the nature of the spacer between the head groups was observed.

Two tetrasiloxane-based imidazolium gemini surfactants with methylene spacer groups (Figure 5b) were synthesized and characterized in [118]. The results showed that with an increase in the length of the spacer, the ability of the surfactants to form micelles increases, while the effectiveness of reducing the surface tension decreases.

A series of organosilicon bis-quaternary amphiphiles was prepared in [119]. It was shown that with an increase in the length of the siloxane spacer, the CMC decreases, and the adsorption characteristics improve. The antimicrobial activity of these compounds was also characterized. Other three cationic silicone surfactants with chloride counterions (Figure 6a) were synthesized in [120]. It was shown that all obtained surfactants have a high surface activity and at the same time a low degree of counterion binding. In [121] pyridine- (Figure 6b) and piperidine-based (Figure 6c) silicone surfactants were synthesized and their properties were investigated. The effect of inorganic salts (NaCl, NaBr, NaI, Na_2_SO_4_) on aggregation of these surfactants was studied.

The authors of [122] obtained three cationic silicone-based surfactants with pyridinium and ammonium fragments and studied the influence of steric factors on the surface activity, adsorption at the interface, thermodynamic characteristics, and aggregation of these amphiphiles. In another work, a series of cationic silicone-based gemini surfactants containing hydrophobic substituents of various lengths (C_n_H_2n+1_, *n* = 8,10,12,14,16,18) was obtained in [123]. It was shown that the morphology of aggregates changes from spherical to rod-like, dumbbell-like and finally string dumbbell-like associates with the increase of surfactant concentration (Figure 7).

In [124] a series of functional silicone-based surfactants were obtained and the solubilizing ability of these amphiphiles with respect to the disperse dye C.I. Disperse Blue 56 was studied. Two new silicone-based carboxylate surfactants with various hydrophobic silicone-containing fragments were obtained in [125]. Both surfactants showed a strong decrease in the surface tension of water. TEM results showed that these amphiphiles formed spherical aggregates above their CMC.

A four-step synthesis and characterization of a new silicone-containing polyurethane surfactant are given in [126]. The resulting polymers have high molecular weight and exhibit excellent surface-active properties. New silicone-containing surfactants based on polyesters with various hydrophobic groups were synthesized and characterized in [127]. The presence of a methylene group between the silicone atoms increases the Lifshitz–Van der Waals forces of the carbosilane hydrophobic groups, thereby increasing the CMC value and the area occupied by one surfactant molecule. Degradable and thermosensitive microgels based on a three-component system containing a silicone component were obtained in [128]. They were of spherical shape with a narrow size distribution and stable in an acidic buffer solution, but decomposed in neutral and base solutions, and capable of loading and releasing doxorubicin (DOX).

The above literature survey demonstrates that the synthesis of novel amphiphilic compounds and investigation of their properties and practical applications have attracted much attention from researchers. Many efforts have been directed on the improvement of the surface activity and adsorption characteristics of surfactants and solubilization capacity of their assemblies, since these are basic properties responsible for the bio- and nanotechnological potential of supramolecular systems. In the case of surfactants with natural or cleavable fragments special attention has been paid to their applications in pharmacy and medicine due to their reduced toxicity, biocompatibility and diverse morphology, with most interest being attracted by cationic surfactants. These amphiphilic compounds are further considered as building blocks and modifiers for nanocarriers capable of enhancing their efficacy as drug delivery systems.

## 2. Self-Assembled Amphiphilic Systems as Smart Drug Nanocarriers

As illustrated above, amphiphilic systems are of particular interest in biomedical applications [2], which can be supported by their use as hydrotropic additives [129], nanocontainers for poorly soluble drugs [130,131,132,133], antimicrobial preparations [134,135], complexing agents for negatively charged biomolecules such as DNA and BSA [136,137,138,139,140], etc. They have been successfully used for the fabrication of drug and gene delivery systems demonstrating numerous advantages compared to non-formulated drugs, i.e., improved bioavailability, higher stability and prolonged circulation in bioenvironment, lower side effects, etc. [141]. Although various types of nanocarriers have been recently designed there are problems preventing efficient transporting of cargo into targeted cells. Among crucial challenges are trafficking in bioenvironment, intracellular uptake and endosomal escape. Therefore, much attention is paid to engineering the nanoparticles addressing these tasks. Some representatives or pioneer examples are summarized in Table 4. 

Importantly, the physicochemical properties, e.g., size, shape, charge characteristics of nanocarriers are key factors determining their biological fate, especially the primary interaction with cell membranes [173,174]. In this context, much effort has been made to design novel effective formulations and their surface functionalization. The size of nanoparticles is documented to affect the formation of corona proteins, with both quantity and quality of protein guided, which essentially controls the opsonization process. Generally, small particles are less opsonized; therefore, they are characterized by longer circulation times and lower accumulation in liver and spleen. Meanwhile the particle shape rather than particle size has a significant impact on the macrophage uptake, biodistribution and targeted delivery. Phagocytosis rates are shown to increase in the series prolate ellipsoid < sphere < oblate ellipsoid [173]. Deviation from spherical shape is responsible for hydrodynamic characteristics, orientation in the blood stream and angle between nanoparticle and membrane, with these factors strongly correlating with trafficking of nanocarriers. Surface charge plays a significant role influencing amount of plasma proteins adsorbed on the nanocarriers, thereby controlling the circulation time. Increase in charge density regardless of the sign of charge results in an increase in opsonization process and decrease in circulation time compared to neutral nanoparticles. Noteworthy, morphology of nanocarriers is significant factor influencing intracellular uptake [175], with the size 50–100 nm considered as optimal for endocytic pathway.

### 2.1. Endosomal Escape

Usually nanoparticles cannot passively pass through the hydrophobic cell membrane. Thus, the cell membrane presents the first barrier to their penetration into the targeted intracellular compartment. At the same time, it is well known that several active intracellular uptake pathways that could be used for nanoparticle internalization are available at the cell membrane. During the first step of intracellular uptake, the cell membrane forms an invagination to take up a portion of extracellular fluid including cargo molecules in it. This active intracellular transport, which is called endocytosis, for cargos smaller than 200 nm can be basically classified into three major categories: receptor-mediated clathrin-dependent endocytosis, caveolar pathway, and the less understood panoply of clathrin- and caveolae-independent pathways [176].

Receptor-mediated clathrin-dependent endocytosis is the mechanism used to internalize cargo into the cells via binding to their specific receptors on the surface of the cell membrane. After formation of these ligand-receptor complexes, the cytosolic protein named clathrin-1 polymerizes on the inner surface of cell membrane and the cargo is engulfed through the formation of clathrin-coated vesicles.

Caveolae-mediated endocytosis is the mechanism of active intracellular transport which involves membrane invaginations called caveolae. Caveolae are formed by assembly of membrane proteins caveolins. It is important to note that endocytosis via caveolae remains poorly characterized, and it is not clear whether caveolae-mediated endocytosis is constitutive, as clathrin-dependent pathway.

There are also a number of pathways for endocytosis, where specific proteins that coat vesicles, such as clathrin or caveolins, were not identified. It is known that endocytosis occurs in cells, when clathrin-dependent endocytosis was inhibited by hypotonic shock, cytosolic potassium depletion, or acidification of cytosol [177]. However, it is not clear under which physiologically normal conditions clathrin- or caveolin-independent endocytosis occurs. Thus, this pathway of internalization is still far from being understood and requires further research to be used for targeted delivery systems.

This information about the main pathways for endocytosis of nanoparticles has been successfully used to develop new nanocarriers for targeted delivery of drugs. For example, mechanisms of receptor-mediated endocytosis can be used for targeted delivery of anticancer drugs. It has been shown that density of certain receptors that are internalized via clathrin-dependent endocytosis is much higher in cancer cells than in normal cells. Functionalization of nanoparticles by the ligands to these receptors can promote both specific binding to the particular cancer cells as well as the intracellular uptake of nanoparticles. The commonly used ligands for anticancer targeted drug delivery are transferrin receptor ligands [178], folate receptor [179], EGFR [180]—just to name a few.

Following internalization, the next crucial step for targeted drug delivery is the intracellular trafficking for localization in the target cellular compartment. It is important to note that in most cases, initially, the intracellular localization of the nanoparticles is not optimal. After internalization of nanoparticles via clathrin-dependent endocytosis or clathrin- and caveolin-independent endocytosis, they will first fuse with early endosomes, which will transform into late endosomes upon the maturation. The late endosomes will then fuse with lysosomes and then nanoparticles will be digested by the hydrolytic enzymes activated at acidic pH occurring in these microdomains. Thus, to reach the targeted intracellular compartment, nanoparticles must be able to escape this endosome-lysosome pathway, bypassing the degradation process. 

Over the past two decades, to prevent the endosomal degradation different strategies have been developed. It was shown that endosomal escape of nanoparticles into the cytoplasm can be promoted by the incorporation of cell membrane penetrating peptides [181] or pH-dependent fusogenic agents into the nanocarrier [182] (Figure 8). Meanwhile, these and other effective approaches have been generalized in the recent literature [156,157]. One of the effective ways for facilitating endosomal escape is the use of endosomolytic agents, that can mediate endosomal escape due to fusion or destabilization of endosomal membrane. These agents may be exemplified by peptides of both anionic and cationic nature. Anionic peptide agents can be designed by the introduction of amino acid residue i.e., aspartic acid and glutamatic acid to the peptide sequence. Due to combination of amino- and carboxylic groups these molecules demonstrate pH dependent behavior, being positively charged under acidic conditions and negatively charged under basic conditions. Cationic endosomolytic agents are designed by modifying peptides with arginine, lysine or histidine moieties, which are positively charged at both physiological and acidic conditions or undergo charge transition upon the lowering in pH (e.g., histidine). These pH-dependent agents demonstrate fusogenic behavior in acidic microenvironment due to electrostatic interaction with negatively charged membranes, inducing membrane fusion and endosomal escape. 

For lipid formulations such as liposomes, modification with cationic or fusogenic lipids has been applied. Cationic lipids destabilize negatively charged endosomal membranes through ion-pairing mechanism, which involves changes in geometry of amphiphilic matrix from cylinder to reverse cone, thereby inducing the formation of reverse hexagonal phase [156]. Similar strategy can be exemplified by the construction of liposomal formulations bearing dioleoylphosphatidyl-ethanolamine and cholesterol as helper lipids, that are capable of changing the surface curvature of membranes and initiate structural rearrangement from liquid crystalline lamellar phase to reverse hexagonal packing under acidic conditions, thereby destabilizing endosomal membranes and allowing payloads to reach the cytosol [157,183]. Effective way is the conjugation of lipid molecules with histidine moieties capable of charge transition behavior. 

Mechanisms of endosomal escape mentioned above are postulated for other organic nanoparticles, including those based on polymeric building blocks. For polymers (e.g., PEI, chitosan, PAMAM dendrimer) bearing ionogenic groups capable of being protonated at acidic pH, additional mechanism is assumed referred to as proton-sponge effect. Generation of positively charged centers in macromolecules is accompanied by accumulation of counterions within the endosomal compartment, thereby inducing rupture of biomembranes due to increase in osmotic pressure and swelling effect [157,158]. Meanwhile, alternative explanation is that pore formation rather than proton-sponge mechanism is responsible for the delivery of drug from endosomal compartment to cytosol by carriers showing enhanced buffering capacity. One more interpretation of the endosomolytic action of polymers bearing aminogroups is the so-called umbrella hypothesis. It treats the swelling effect and “osmotic shock” in terms of unbending the polymer chains due to electrostatic repulsion of protonated amino groups [183]. The role of functionalization of PEI affecting its pK value, concentration-dependent effect and insistence of balance between efficacy in endosomal escape and cytotoxicity of formulations at high PEI content are emphasized. Mechanism of rupture of endosomal membrane analogous to the proton-sponge effect is ascribed to lysosome tropic agents, such as primary amines, chloroquines, primaquine, ammonium chloride [183]. These additives are assumed to decrease activity of lysosome enzymes, thereby allowing nanocarriers to release to cytosol.

Development of nanocarriers that use caveolae-mediated endocytosis for their internalization also can be considered as a perspective approach to escape from endosome-lysosome pathway. It was shown that caveolae can fuse with early endosome but also with “caveosome”. Caveosomes are the cell compartments that exist at neutral pH and do not have hydrolytic enzymes [176]. Thus, nanoparticles taking this pathway to a certain extent also avoid a fate of liposomal degradation.

It should be noted that not only the endo-lysosomal system affects the nanoparticle but also vice versa. In recent studies, it was shown that cells treated with nanosized fluvastatin show increasing levels of autophagosome formation that competitively inhibit lysosomal activity [184]. It was shown that after internalization of nanoparticles functionalized with cell membrane penetrating peptide (TAT peptide), bystander nanoparticles, without any cell-penetrating ligands, can enter into the cells through the same endocytic pathway as functionalized nanoparticles [185]. It also has been shown that nanoparticles covered with a mix of positively and negatively charged ligands aggregate into supraparticles in the lysosomes of cancer cells, disrupting the integrity and killing the cells without involving any anticancer drugs [186].

### 2.2. Liposomal Nanocarriers

#### 2.2.1. Advantages and Limitations of Liposomal Formulations

Table 4 summarizes selected examples of solving the known problems associated with the drug delivery systems on their way towards clinical application. This is very incomplete list limited by the choice of authors and the scope of the review. Meanwhile, most of the examples of the challenge overcoming given in Table 4 are representative and may be attributed to a wide variety of nanocarriers. The latter is demonstrated on the strategies developed for solving the endosomal escape problem. At the same time, essential part of Table 4 is devoted to liposomal formulation that are the mostly well on the track of achieving the practical application. Therefore, these nanocarriers are the most relevant cases for analysis of strength versus weakness of nanocarriers and choosing the strategies for their improvement.

Among the clinically relevant nanocarriers, liposomes have a special place, since only liposomal formulations of antitumor drugs were first approved for clinical practice. This can be exemplified by liposomal DOX preparation, Doxil, which was the first liposomal drug approved by FDA, which was followed by the liposomal vincristine carrier Marqibo and liposomal daunorubicin drug DaunoXome, etc. [187,188,189,190]. Such incremental advance of liposomal nanocarriers is based on their beneficial structural, physicochemical and pharmaceutical characteristics. The lipid-based composition of liposomes is responsible for their biocompatibility and low toxicity; vesicle-like structure of liposomes composed of aqueous lumen and lipophilic periphery provides conditions for incapsulating both hydrophilic and lipophilic drugs, with their incorporation into liposomes improving their solubility and bioavailability; protecting them from biodegradation, enhancing the cellular uptake, etc.

In the history of liposomal drug delivery systems, a lot of ways of improving therapeutic efficacy of drugs have been found [188,191], as exemplified in the upper rows of Table 4. The first generation of liposomes composed of phospholipids and cholesterol (the so-called conventional liposomes) allowed to significantly improve the therapeutic index of loaded drugs, e.g., DOX and amphotericin due to affecting their pharmacokinetics, biodistribution and in vivo toxicity. However, these conventional liposomes underwent fast clearance by the reticuloendothelial system (RES) due to opsonization by plasma proteins and uptake by macrophages. The liver and spleen were found to accumulate the largest fraction of liposomes. Eventually, the so-called stealth liposomes, modified with polyethylene glycol, were the first formulations to reach the commercial market for cancer therapy. They were the solution of fast clearance problem by the RES due to reduced opsonization of the highly hydrated PEG-decorated nanocarriers, which hinder their recognition by macrophages. Along with enhanced circulation time, these nanocarriers were found to accumulate in the tumor tissue that is more vulnerable for nanoparticle accumulation than healthy tissues. This phenomenon is referred to as enhanced permeability and retention (EPR) effect [192].

In turn, pegylated liposomes along with the stabilization effect on nanoparticles resulted in a body of concerns referred to as the PEG dilemma, with limitations on cellular uptake and further endosomal escape, as well as accelerated blood clearance effect (ABC) occurred. The pegylation significantly lowered the liposomes’ interaction with the target tissue alongside opsonins. To overcome this problem, the next generation of liposomal formulations were developed [193,194]. Among different approaches to improve the cellular absorption of formulated drugs, the design of ligand-mediated nanocarriers with target molecules is being implemented. For this purpose, different peptides are used complementary to such receptors as integrin, transferrin, human epidermal growth factor 2 (HER2) and others. In the middle of the range of peptide ligands, is Arg-Gly-Asp (RGD), one of the most popular motifs used for targeted delivery to tumor cells overexpressing integrin. A recombinant monoclonal antibody trastuzumab was used as a HER2-specific drug. The so-called immunoliposomes decorated with the antibody vectors have received much attention in recent literature as an improved generation of liposomal carriers [193,194]. Pegylation of nanocarriers is documented to inhibit endosomal escape, which was testified in the transfection efficacy studies. Therefore, unique approaches were needed to answer this part of the PEG dilemma, including design of pH-responsive PEG-lipids or PEG-PEI conjugates, cleavable pegylated agents bridged by pH-sensitive bonds, e.g., ester, amide, disulfide, that undergo cleavage at low pH with the PEG residue released [184,193,194]. 

Concerning the different aspects of incremental advances in design and practical application of liposomal nanocarriers, a variety of liposomal drugs are on the market, and much more are in different stages of clinical trials, which is comprehensively discussed in reviews [188,189,190]. This provides evidence that all the problems occurring with liposomes can be solved without going beyond the liposomes. Meanwhile alternative nanocarriers aimed at the overcoming the limitations of liposomes are being developed. Some of such cases are given in Table 4 and exemplified by lipid formulations, e.g., solid lipid nanoparticles, vesicle-like carriers, e.g., niosomes and hybrid formulations, cerasomes [195,196,197]. These nanocarriers tend to further improve biocompatibility, loading capacity, targeting efficiency, circulation time, and morphological stability by combining e.g., the lipid/nonionic surfactant or liposomal/silica nanoparticles features. Polymeric nanocarriers with mucoadhesive characteristics can be alternatively involved in the case of oral administration of drugs [159]. Nevertheless, we would like to adhere to the point of view emphasizing that main problems responsible for the bottleneck in the translation of liposomes to the clinical practice are common obstacles that can be faced by the majority of nanocarriers rather than liposomal formulations only [188]. They are basically related to manufacturing process and conjugated with the loss of stability and quality of product upon scaling of the preparation procedure, reliability and reproducibility of final drug, and regulation of manufacturing process by government, with intellectual property intricacies involved. The majority of the problems arising on the stage of manufacturing process are related to the increasing complexity of the composition of nanocontainers, with multistage chemical and biochemical synthesis being involved. The possibility to alleviate these problems on the research stage is probably to move towards the reduction of the complexity of chemical modification of nanocarriers and contribution of the synthetic manipulations. Therefore, we focus on the noncovalent modification of nanocarriers, with no complex synthesis involved. With this in mind we discuss below selected aspects related to cationic liposomes (CLs), with the strategies of noncovalent surface modification of liposomes by cationic surfactants being emphasized.

#### 2.2.2. Cationic Liposomes as Drug Carriers

A survey of recent literature allows us to provide a brief outline of the urgency of the studies focusing on CLs. One of the most studied and recognized sphere of CLs application is their use for transfection. Their positive charge makes them an exceptional nanocarrier for negatively charged nucleic acid sequences and is advantageous due to higher tissue uptake and endosomal escape capabilities [198]. For a few decades, cationic liposomes have attracted attention due to their lower immunogenicity, little to no toxicity, when biodegradable lipids are used, and they have since been studied thoroughly to be used in transfection [199]. Thus, scientific interest in CLs for transfection has not dwindled in the past few years [198,200,201,202,203,204]. Another major field of CL application is their use as vaccine adjuvants due to their ability to protect the delivered vaccine, retain it at the injection site, and some degree of immunostimulation [205,206].

One of the well-known downsides of the CLs is their toxicity, which is inherent to the positive charge; thoughtful dosage can mitigate the toxic effects, as shown in the recent work by Knudsen and co-workers [207]. Another major downside is rapid clearance from the bloodstream, resulting from the liposome charge [208]. Given the flexible nature of liposomes, it should be possible to avoid these obstacles, as we will see in the following paragraphs.

Meanwhile, attempts are being made to investigate CLs as delivery vehicles. The cationic surface charge is very beneficial when dealing with endosomal trafficking due to the negative charge of endosome membrane [147,209]. Trementozzi et al. have characterized a ternary lipid composition that is able to release the contents in a triggerable manner, and addition of only 3 mol % of 1,2-dioleoyl-3-trimethylammonium propane to promote liposome-endosome affinity and fusion has shown a 5-fold increase of formulation cytotoxicity [210]. Another use of CLs, that has been gaining popularity, is co-delivery of anticancer agents and nucleic acids, since CLs are the only type of liposomes that are able to encapsulate both drugs and negatively charged nucleotides well. Some anticancer agents, such as paclitaxel are object to the development of chemoresistance based on survivin overexpression in tumor cells. Other types of cancer cells overexpress their essential proteins that provide proliferation, each of which could be precisely controlled with siRNA. Cationic liposomes are the only tool that can perform well delivering both an anticancer agent and a siRNA fragment, thus providing a more complex and efficient approach to tumor treatment [201,204]. 

To summarize this brief discussion, cationic liposomes are of current importance in view of their high biomedical potentiality. Nevertheless, in some cases large positive charge of nanocarriers may prevent their efficacy due to, e.g., non-specific membrane fusion with other than endosomal membranes, enhanced opsonization or agglutination effects, elevated toxicity of formulations, etc. Therefore, strategies of fabrication of the partially charged liposomes and/or charge-switchable nanocarriers are of importance. 

#### 2.2.3. Cationic Liposomes Obtained by Non-Covalent Modification with Surfactants

As an alternative to the formation of mixed compositions based on phosphatidylcholines and cationic lipids, a strategy of non-covalent modification of the surface of liposomes by cationic surfactants is actively developing (Figure 9). Among the advantages of this approach, the following can be noted: (1) commercial availability of cationic surfactants compared to cationic lipids; (2) the structural diversity of cationic surfactants with the ability to embed targeted or functional fragments that enhance binding; (3) fine-tuning the properties of liposomes (substrate release rate, encapsulation and loading efficiency) depending on the nature of the surfactant. Cationic liposomes due to their charge help solve the following problems: passage through the blood-brain barrier (BBB) (Figure 10); delivery of antineoplastic agents by increasing cellular uptake; transdermal drug delivery [211,212,213,214,215].

In the formation of modified liposomes, the degree of oligomerization of a cationic surfactant (monomeric, gemini or trimeric) and the ratio of lipid to surfactant are crucial factors, since only at a low surfactant concentration the incorporation of surfactant into the lipid bilayer of liposomes is possible, while after a certain threshold value complete destruction of the lipid bilayer and the formation of mixed surfactant-lipid micelles are observed [217,218,219]. The structure of the head group of surfactants can be different.

We begin our consideration of this issue with monomeric single-tailed surfactants (Figure 11). Pegylated cationic liposomes modified with a new arginine surfactant were described in [220], which were used to increase the bioavailability of new 2-benzimidazolylquinoxalines derivatives poorly soluble in water and showed highly selective cytotoxicity against a cervical cancer cell line (HeLa). Another benzimidazole derivative, 7-(benzimidazol-2-yl)-6-(2,4-dichlorophenyl)-2-thioxo-2,3-dihydropteridin-4(1*H*)-one, when encapsulated in imidazolium surfactant-modified liposomes shows a sharp decrease in viability of diseased cells (HeLa) and increase of viability of normal cells (Chang liver) [221]. Cationic liposomes based on 1,2-dipalmitoyl-sn-glycero-3-phosphocholine (DPPC) and alkyltriphenylphosphonium bromides were obtained by non-covalent surface modification with amphiphiles using varying hydrocarbon tail lengths and lipid/surfactant ratio and showed a high degree of colocalization with mitochondria of human pancreatic cancer cell line (PANC-1) [222]. Hybrid liposomes based on DPPC and 1-methyl-3-alkylimidazolium bromides have been used for metronidazole encapsulation [131]. It was shown that the highest value of encapsulation efficiency, equal to 75%, is achieved in the case of surface modification with an octadecyl derivative. New sterically hindered phenol containing a quaternary ammonium moiety (SHP-n-Q) was also used for incorporation into liposomes [223]. Mixed liposomes based on L-α-phosphatidylcholine and SHP-n-Q showed long-term stability (1 year) and prolonged release of rhodamine B. A cationic surfactant, N,N,N-triethyl-N-(12-naphthoxydodecyl)ammonium, was inserted in liposomes based on phosphatidylcholine [224]. Modified liposomes loaded with cefepime antibiotic can be used in bacterial infections against *Escherichia coli* with high inhibitory activity. The investigation on liposomes formulated with 1,2-dimyristoyl-sn-glycero-3-phosphatidylcholine (DMPC) and cationic surfactants: (1*S*, 2*S*)-N-hexadecyl-N-methylprolinolinium bromide, (1*R*, 2*S*)-N-hexadecyl-N-methylprolinolinium bromide, N-hexadecyl-L-prolinolinium hydrobromide, N,N-dihexadecyl-L-prolinolinium bromide, in the presence and in the absence of *m*-tetra-hydroxyphenylchlorin is presented in [225]. Different liposome formulations have been evaluated for use against methicillin-resistant *Staphylococcus aureus* bacteria and one of them has shown results comparable to free chlorin. The traditional cationic surfactant CTAB was used in the work to impart a positive charge to niosomes based on N-myristoylserinol and cholesterol [226]. In the increasing presence of CTAB, the size of niosomes decreases from 300 to 85 nm, which the authors attribute to the electrostatic repulsion of particles, and the fluidity of liposomes.

Very often, long-chain amines are used to modify the surface of liposomes, especially when developing compositions for treating skin diseases. Hexadecylamine was integrated into the liposomes based on DPPC and cholesterol (8:2 (mol)) obtained by microfluidic assembly under conditions of optimization of flow rate to increase colloidal stability [227]. Additionally, a biodegradable anionic pectin polymer was applied to the surface of the modified liposomes. However, it was not possible to load branched-chain amino acids (leucine, isoleucine, and valine) into cationic liposomes. Liposomes of the composition phosphatidylcholine, cholesterol, stearylamine, and Tween 20 were obtained for transdermal delivery of imperatorin (a Chinese herbal medicine) [228]. Compared to liposomes with a different surface charge, cationic liposomes have been shown to be more effective in models of skin inflammations or bacterial infections. The authors suggest cationic liposomes delivered the highest amount of the active component across the skin into deeper regions of the epidermis [228]. The authors of another article also used stearylamine to produce deformable cationic liposomes to increase the bioavailability of curcumin [229]. The formed system was non-toxic and safe for skin use. It was found that cationic liposomes significantly inhibit bacterial *Staphylococcus aureus* and *Streptococcus pyogenes* growth in vitro. Most persistent penetration of human skin ex vivo was achieved with use of Curcumin-CLs.

Between monomeric and gemini surfactants, an intermediate position is occupied by surfactants with two hydrophobic tails and one head group, which can form vesicles instead of micelles in a solution and have an ability to integrate into a lipid bilayer (Figure 12). For example, mixed cationic liposomes based on L-α-phosphatidylcholine and dihexadecylmethylhydroxy-ethylammonium bromide were obtained in [230] with varying lipid/surfactant ratio. When the liposomes loaded with 2-PAM were intranasally administered, reactivation of brain acetylcholinesterase was observed at 12% for rats subjected to paraoxon poisoning. Liposomes based on egg phosphatidylcholine, cholesterol, Tween 80 and didecyldimethylammonium bromide were tested to increase the penetration of andrographolide into the brain [231] on artificial membranes (PAMPA) and hCMEC/D3 cells. The paper discusses the mechanism of nanoparticle permeation through the hCMEC/D3 monolayer, highlighting caveolae-mediated endocytosis as the main one. The authors of [232] found that the presence of dimethyldioctadecylammonium bromide in the lipid bilayer of DPPC leads to an increase in nanoparticle stiffness, which affects the penetration through the skin. Apparently, the positive surface charge of the liposomes contributes to their retention in the stratum corneum and upper layers of epidermis. This statement is opposite to the conclusion of the authors of articles [228,229], where cationic liposomes with stearylamine penetrate into the deep layers of the skin. Most likely, the structure of the surfactant has a decisive influence on the rigidity of the lipid bilayer.

The next interesting block about modified liposomes is associated with gemini surfactants (Figure 13). Modification of liposomes by new 1,4-diazabicyclo[2.2.2]octane surfactants with four charges and two hydrophobic chains (n-DABCO-s-DABCO-n, where s = 2, 6, 12 and n = 12, 14, 16, 18) was carried out in [233].

The liposomal formulation based on phosphatidylcholine and the tetradecyl gemini derivative DABCO showed the highest antibacterial activity against *Staphylococcus aureus* (MIC = 7.8 μg·mL^−1^) and lowest cytotoxicity (IC_50_ > 125). The resulting composition was tested for transdermal drug delivery. Also, for transdermal delivery, niosomes were obtained using lysine-based gemini surfactant and cholesterol [234]. The most appropriate administration route and formulation are to be used to achieve best pharmacological results.

Interactions of surfactant modified liposomes with plasma membranes are elucidated in [235,236], where authors study how liposomes based on DMPC and a gemini surfactant ((2S,3S)-2,3-dimethoxy-1,4-bis(N-hexadecyl-N,N-dimethylammonium)butane bromide) are internalized by cells, using confocal microscopy and electro-rotation. The investigation on the physicochemical properties of liposomes formulated with a DPPC or 1,2-dioleoyl-sn-glycero-3-phosphocholine (DOPC) [237], cholesterol, and diastereomeric gemini amphiphiles (2S,3S)-2,3-dimethoxy-1,4-bis(N-hexadecyl-N,N-dimethylammonium) butane bromide or (*S,R*)-2,3-dimethoxy-1,4-bis(N-hexadecyl-N,N-dimethyl- ammonio)butane bromide and on their ability to deliver the plant alkaloid voacamine to U-2 OS/DX osteosarcoma cells was carried out. The effect of incorporating surfactants into the lipid bilayer on the efficiency of encapsulation depends on the nature of the lipid itself. A series of papers by this team of authors ends up in preparation of pegylated gemini-modified liposomes [238].

In [219] the authors have studied how lysine spacer length and aggregate structure of peptide gemini surfactants affect the interactions of the surfactants with DOPC, 1,2-dioleoyl-sn-glycero-3-phospho-rac-(1-glycerol) sodium salt (DOPG) and DOPC/DOPG (1:1) vesicles. Derivative with butyl spacers were found to be optimal in terms of balance of electrostatic, hydrophobic and hydrogen bonding interactions and the most interacting with the phospholipid bilayer.

Thus, according to the increasing number of articles devoted to the production of cationic liposomes by non-covalent surface modification with surfactants, we can say that the technique is successful and relevant. By selecting the structure of the lipid and surfactant, the ratio of the components, it is possible to control the efficiency of encapsulation and loading, the release rate of the substrate, the ability of the liposomal formulation to overcome biological barriers (cancer cell membrane, BBB, various skin levels) to obtain commercially available nanocontainers. 

#### 2.2.4. pH-Dependent Cationic Lipids

pH-responsive liposome formulations present a very vast range of benefits and have been studied since 1980s and especially in the last decades as candidates for second-generation drug delivery nanoparticles, but they are yet to be approved for clinical applications [239]. One way of utilizing pH sensitivity is to craft lipids that destabilize the liposome membrane when exposed to acidic environment, which can be found in the vicinity of tumor tissue. This concept was demonstrated in 1980 by Yatvin et al. [240]. Later it was concluded that acidity can also induce fusion in pH-sensitive liposomes, which can further contribute to the release of a substrate [241]. Such responsive liposomes remain an object of interest and are being studied with application of novel pH-sensitive lipids [242,243].

Another unique feature of such nanocontainers can help solve the problem of premature clearance from bloodstream [208], pH-responsive lipids and surfactants can be applied to allow vesicles that appear almost neutral in the bloodstream, but obtain a positive charge when exposed to lower pH environments like such in the endosomes [147]. This approach is based on protonation of the 3ß-[N-(N’,N’-dimethylaminoethane)carbamoyl]cholesterol hydrochloride (DC–cholesterol) lipid in acidic media, which is common for endosome lumen, that in turn leads to liposome fusion with the endosome membrane and contributes to the endosomal escape. This result received confirmation by comparison of colocalization efficacy of DC-cholesterol liposomes in the early and late lysosomes. A different approach to utilize pH-responsiveness involves thermally treated fumed silica, silanol groups of which are protonated in acidic media and lead to liposome collapse and rapid release of attached substrate, as described in [148]. One more way of exploiting in vitro and in vivo pH differences is attachment of cleavable PEG chains to the liposomes, that protect the nanocarriers from quick reticuloendothelial uptake, but get cut off in the acidic media of tumors or some intracellular regions (Figure 14). Vinyl ether, hydrazine, ester, amide, imide, peptide, disulfide bonds are potential candidates for crafting of cleavable PEG-conjugated lipids [146,193]. This approach also helps to avoid targeting peptide degradation in circulation by exposing the depegylated liposome membrane only in the tumor area [146]. This is one of the most promising ways to actively target tumors due to the combination of enhanced permeability and retention effect and acidic extracellular pH of cancerous tissues [244,245]. In the last decade, the researchers find more and more intelligent ways of exploiting acidic media in bioenvironments, which must lead to the development of novel nanocarriers, that simultaneously have strong sides of pegylated and cationic liposomes.

## 3. Mucoadhesive Properties of Supramolecular Systems Based on Macrocycles and Polymers

In the above discussion, we focus on the beneficial applications of nanocarriers bearing cationic charges, focusing on the family of lipid formulations modified with surfactants. At the same time, in the field of organic nanoparticles as drug delivery systems (nanomedicines) the special class of biomimetic transporters of macrocyclic nature and polymeric nanocarriers, especially those of cationic nature, play a significant role which remained unaddressed in the review. Therefore, the final part focuses on supramolecular assemblies based on mixed macrocyclic and polymeric systems, with special attention paid to cationic polymers due to their enhanced mucoadhesive properties. Among the various methods of delivery and modification of dosage forms, the phenomenon of mucoadhesion, i.e., the ability of the compositions to adhere to the surface of the mucous membrane (mucose) is of particular importance for increasing the drug bioavailability. In order to keep the encapsulated drug in the tissues of the body (in the mucous membranes of the innards, digestive tract, respiratory system, urogenital tract) for a long period of time, the drug nanocontainer should have mucoadhesive properties. The lack of adherence to these tissues is the main problem observed in clinical trials with drugs. Hence, the understanding of the mechanisms of mucoadhesion with mucus is of paramount importance for creating effective systems for the delivery of drugs to the human body. In this line of research, special attention is paid to the modification of systems by water-soluble and water-swelling polymers, which can improve mucoadhesive properties in order to increase the therapeutic effect [246]. In this section, various types of mucoadhesive systems based on macrocycles and polymers will be considered, with their combination achieving the synergetic effect on solubility, bioavailability, targeting and mucoadhesiveness (Figure 15).

It is well known that cyclodextrins (CDs) are able to increase the solubility of various lipophilic molecules by forming relatively stable inclusion complexes. The stability of these complexes is usually higher, when the size of the guest molecule is suitable for the size of the macrocycle cavity. Due to the receptor action and low toxicity, CDs are widely used in the design of drug delivery systems. Chemical modification of CDs allows the formation of derivatives with improved physicochemical properties (good solubility, optimal size and ionic charge) as compared to natural CDs, which can positively affect their biomedical efficiency [247]. This modification makes it possible to include a wide range of hydrophobic, hydrophilic or amphiphilic fragments, particularly of a polymer nature, in the macrocyclic framework.

The formation of inclusion complexes of CDs accompanied by an increase in the aqueous solubility of drugs can lead to spontaneous self-assembly of these complexes. Encapsulation of the drug telmisartan in the γ-CD cavity can initiate the aggregation of inclusion complexes. The γ-CD complexes with telmisartan are able to aggregate into spherical nanoparticles with diameters from 50 nm to 500 nm [160]. However, these aggregates are metastable and are likely to precipitate. In order to maintain the stability of the aqueous composition of eye drops, hydroxypropyl methylcellulose was added to the obtained complexes in two ways: polymer was added (i) during the preparation of the telmisartan/γ-CD powder to form a ternary complex and (ii) to the environment of the formed complex to form a binary complex. The ternary complex exhibits worse mucoadhesive properties compared to the binary system, but the ternary complex was more stable, since when stored at 4 °C for 6 months, there was no noticeable change in the average particle size (2.0–2.5 μm).

Hydroxypropyl methylcellulose and other water-soluble polymers (chitosan and hyaluronic acid) can be used not only for stabilization, but also to increase the solubilization of the drug as an additional agent to CDs. The ternary CD–polymer–celecoxib complexes form joint aggregates ranging from 250 to 350 nm in size and display a large solubilization capacity for the drug [161]. Eye drops of microsuspension containing ternary aggregates based on randomly methylated β-CD–hyaluronic acid–celecoxib complexes showed relatively good mucoadhesion and cytocompatibility with the human retinal pigment epithelial cell line [162]. Another work [163] demonstrated that the complexation of piroxicam with CDs can be used to ensure controlled in vitro release of the drug and increase its penetration through the buccal mucosa, and the presence of chitosan in these complexes significantly increases the transport properties of the drug. Ahmed et al. also showed that the presence of an increased amount of CDs in the mucoadhesive system with a polymer increases effectively the solubility of poorly soluble drug as well as improves absorption and penetration of the drug through the intestinal mucosa [164].

In [248] the ability of a conjugate based on chitosan glycol and β-CD as a container of hydrophobic antitumor drugs (5-fluorouracil, DOX and vinblastine) was evaluated. This drug nanocontainer has both mucoadhesive properties due to the polymer and good encapsulating properties due to the macrocycle. It is also possible to create a pH-sensitive carrier of antitumor drugs from chitosan glycol and carboxymethyl-β-CD [249]. Due to the mucoadhesive property of chitosan glycol, this conjugate can interact with porcine stomach mucin, and the CD fragment having several free carboxylic acids groups can bind DOX in weakly basic medium (pH 7.4) and release it in acidic medium (pH 5.0).

Obviously, the introduction of CD fragments into the polymer chain of chitosan can lead to the formation of a molecular carrier with encapsulating properties for drugs, transport properties through biological barriers and the ability to control drug release. However, it is not always possible to solve the problem of the low solubility of chitosan in water by crosslinking of CD with the polymer; therefore, the chitosan was quaternized [165,250]. Conjugates based on β-CD and quaternized chitosan can aggregate into large particles ranging in size from 800 to 3000 nm, where β-CD is located inside the hydrophobic core and quaternary ammonium fragments are located outside [251]. These aggregates can solubilize eugenol, exhibit mucoadhesive properties and antimicrobial activity during drug delivery through the mucous membrane. It is interesting to note that the release of eugenol from aggregates was prolonged due to aggregation ability, in contrast to the conjugate, where the macrocyclic and polymer components are crosslinked with a citric acid spacer. However, the mucoadhesiveness of the latter was revealed mainly due to the electrostatic interaction between the positively charged amino groups of chitosan and negatively charged mucous membranes, as well as the hydrogen bond between the carboxyl and hydroxyl groups of citrate spacers and mucus glycoprotein. Moreover, a cytotoxicity test showed that chitosan–CD conjugates revealed less toxicity towards buccal mucosal cells than the initial quaternized chitosan [252].

Polymeric surfactants may also be involved in the development of mucoadhesive systems. Pluronics were used to improve the delivery of itraconazole to the eyes [253]. The surfactant mixture of two Pluronics (F127 and F68) provides the hydrophobic core needed to capture itraconazole. However, there is a likelihood of corneal tissue irritation when using these surfactants in relatively high concentrations. Micellar dispersion formed from synperonic surfactants, polyethylene oxide and β-CD turned out to be safe, stable, mucoadhesive and effective for penetration of the drug through the rabbit cornea [166]. The sizes of these compositions were in the range of 180 to 360 nm, which is favorable for transcorneal drug penetration through the intercellular pathways [254]. The resulting dispersions demonstrated the rapid rate and extent of distribution of the encapsulated itraconazole in vitro and ex vivo compared to a free drug suspension. In other work [255] the magnetic nanoparticles based on iron oxide were developed by precipitation of iron salts in the presence of ammonia, and four different compositions based on them (particles without coating, with β-CD, with Pluronic 127 and with a mixture of macrocycle and polymer) were created. The latter composition containing particles of about 117 nm in size was found to be effective for binding to mucus compared to other nanoparticles studied, which was evident from the results of ex vivo absorption of these nanoparticles by the gastrointestinal tract, ovaries, pancreas, and colon.

The combination of hydroxypropyl-β-CD with Tween 80 makes it possible to stabilize the drug econazole nitrate, protecting the encapsulated drug from aggregation for one year at room temperature [167]. This composition was suspended in chitosan acidic solution to increase the release and bioavailability of the drug. An in vivo evaluation in eyes of an albino rabbit demonstrated a higher bioavailability of the selected drug suspended in chitosan compared to similar drug suspended in isotonic buffer at pH 7.4. 

Hydroxypropyl-β-CD and polyvinylpyrrolidone were also involved in the creation of rapidly dissolving nanofibers capable of encapsulating clotrimazole [256]. These nanofibers showed rapid drug release and antifungal activity, but the treatment of oral candidiasis requires a sufficiently long period of time, during which the concentration of the drug exceeds the MIC. Therefore, to slow down the release of the drug, the nanofibers were coated with additional layers of chitosan-cysteine and polyvinyl alcohol, which also improved mucoadhesive properties [168].

CDs can be used not only to increase the water solubility of drugs, but also to improve their taste. Hydroxypropyl-β-CD eliminates the unpleasant taste of irsogladine maleate used for treatment of stomatitis. Addition of the macrocycle to the mucoadhesive gum ghatti, consisting of high-molecular-weight polysaccharides, leads to the formation of a viscous solution as a result of aggregation due to hydrogen bonds between the macrocycle and the polymer [257].

Mucoadhesive microspheres based on a complex of γ-CD and raloxifene can be obtained using various proportions of carbopol and hydroxypropyl methylcellulose. The size of the obtained particles on the percentage of polymers ranged from 3 to 15 microns, and the encapsulation efficiency of raloxifene ranged from 81.63 to 87.73%. These microspheres had remarkable mucoadhesion and controlled drug release lasting up to 24 h. More than 60% of the oral dose was kept for 4 h in the stomach of Wistar rats [258].

The mixed compositions based on CDs and polymers can be used not only to encapsulate drugs, but also proteins and enzymes. In [259] microspheres consisting of chitosan, hydroxypropyl-β-CD and polyethylene glycol were successfully developed. The obtained microspheres had a diameter of 6–7 μm, and the BSA encapsulation efficiency was more than 70%. The stability of primary, secondary and tertiary structures of bound BSA was confirmed by a set of physicochemical methods. In other work [260] Bruton’s tyrosine kinase inhibitor, ibrutinib, was loaded into nanoparticles based on chitosan and sulfobutyl ether-β-CD due to the inclusive participation of the latter, and the activity of ibrutinib did not disappear. These nanoparticles loaded with ibrutinib had a diameter of 280 nm and a zeta potential of +19.1 mV, which decreased to +9.2 mV after contact with negatively charged sialic acid residues in mucin. In addition, they showed a prolonged and pH-independent release of the drug in the mimic gastrointestinal tract.

Nanoparticles based on hydroxypropyl-β-CD and chitosan can be used in targeting various organs of the gastrointestinal tract. A study of the release of hydrophobic substrate from this composition showed a sharp release (more than 40%) in the gastric fluid due to the breakdown of chitosan and β-CD inclusion complex at low pH [261]. As in the case of hydrophilic substrate, only a small amount of hydrophobic substrate is released in the intestinal fluid. In the colonic fluid, there is a sharp increase in release due to enzymatic cleavage of chitosan, as well as the decomposition of CD molecule by bacterial amylases.

Besides a wide range of different nanoparticles and microspheres, mucoadhesive micro- and nanocapsules can be obtained from macrocycles and polymers. Simvastatin microcapsules were obtained by complexation with hydroxypropyl-β-CD, hydrophilic sodium alginate and mucilage obtained from *Dillenia indica* [262]. Drug release from such microcapsules reached 73% within 12 h, and this delay was due to the *Dillenia* polymer. In addition, *Dillenia* has sufficient mucoadhesive properties and good compatibility with the drug or any other ingredients of the composition.

Amphiphilic CDs and polymers can be used to create mucoadhesive drug nanocontainers. β-CD with hexyl tails forms nanocapsules with poly-ε-caprolactone for encapsulation of camptothecin [169]. Drug loading and cellular interaction of these nanocapsules can be regulated by coating with chitosan. The coating of the chitosan layer on nanoparticles consisting of macrocycle–drug complexes not only increases the drug loading, but also decreases the release of camptothecin. The presence of a positive charge of the obtained nanoparticles provides higher absorption due to electrostatic interaction with negatively charged intestinal lumen epithelial cells.

The addition of hydroxypropyl-β-CD as a permeation enhancer improves the penetration of timolol maleate through the bovine cornea [170]. However, it was found that timolol formulations containing bioadhesive polymers (hyaluronic acid, chitosan, alginate) alone or together with hydroxypropyl-β-CD can reduce drug penetration. When using polymers as stabilizers of complexes between CDs and drugs, it is necessary to take into account the competitive binding of the polymer with the macrocycle. For example, poloxamer 407 can compete with drug molecules to form inclusion complexes with CD [263,264]. 

Non-covalent aggregation of macrocycles with polymers was used to develop mucoadhesive compositions with effective encapsulating ability. Ion cross-linked nanoparticles based on oppositely charged chitosan and sulfobutyl ether-β-CD are an interesting system for the delivery of econazole nitrate to the mucous membrane of the eye [265]. Half of the drug encapsulated in these nanoparticles is released within 8 h. In addition, the nanoparticles showed mucoadhesive properties, which allow them to interact with the mucous membrane of the eye for an extended period of time and thus provide an enhanced and controlled effect of the drug on the surface of rabbit eyes.

The use of complexation between α-CD molecules and polysaccharides modified with hydrophobic alkyl chains leads to the spontaneous formation of nano- or micron-sized platelets that can encapsulate antimicrobials for effective treatment of mucosal infections [266,267]. The crystal platelets have layered structures consisting of complexes of CDs with alkylated guests, such as polymer framework bearing alkyl chains (Figure 16). Chitosan derivatives capable of being located on the surface of aggregates can be used as polymers in the formation of these platelets [268]. The contact area of the platelets with the biological surface is greater compared to spherical aggregates. Therefore, the adhesion force of the platelets is much greater and provides better binding to mucin. The electrostatic forces between chitosan and mucin chains can stretch the outer chitosan layer of the platelets. If the cohesive force of the chitosan inside the platelet is weaker than the cohesive force between chitosan and mucin, the system can break down in the intermediate layer of the platelet. As a result, the surface layer can be torn off from the platelet with the destruction of the latter, that is important when developing systems for targeted drug delivery with controlled release.

In addition to crosslinking of macrocycles on flexible open-chain molecules, β-CD can be used to modify the mesoporous silica nanoparticles with hydroxyl, amino, and thiol groups [171]. Thiol-functionalized nanoparticles exhibit significantly higher urothelial mucoadhesiveness compared to hydroxyl- and amino-functionalized nanoparticles, which is related with the formation of disulfide bonds between thiol functionalized nanoparticles and cysteine subdomains of mucus glycoproteins. Anticancer DOX can be encapsulated in these nanoparticles and slowly released in an acidic environment.

The authors of [172] definitely proved that the mucoadhesion of the nanoparticles of methyl-β-CD in combination with polyisobutyl cyanoacrylate and thiolated chitosan is a mandatory condition to enhance the intestinal permeability of docetaxel. The obtained nanoparticles were spherical, had a diameter in the range from 200 to 400 nm with a positive charge, and the encapsulation efficiency with respect to drug was 70–80%. In vitro experiments on a model intestinal medium showed that the docetaxel is gradually released, reaching 60% of release after 24 h and 100% after 48 h. The intestinal permeability of drug loaded into the nanoparticles was higher than in the control ethanol solution of this drug.

For modification of mucoadhesive nanoparticles not only CDs can be used, but also derivatives of porphyrins, which are a promising choice for drug delivery systems in photodynamic therapy, combining the advantages of efficient drug transport to target cells and reducing of undesirable side effects. The surface of nanoparticles based on biodegradable poly-DL-lactide-co-glycolide and 5,10,15,20-tetrakis(*m*-hydroxyphenyl)porphyrin can be modified with polyethylene glycol or chitosan [269]. The highest number of pegylated nanoparticles was found in human colon adenocarcinoma cells. Therefore, pegylated porphyrin nanoparticles are a promising drug delivery system for topical treatment of gastrointestinal cancer.

Summarizing the results of mucoadhesive study of mixed macrocycle–polymer systems, when designing effective delivery systems for poorly water-soluble drugs, CDs must be added to mucoadhesive polymers to provide good drug solubility in the aqueous medium, adequate bioavailability and desired drug effect. CDs not only increase the solubility of the lipophilic active substances, but also have the potential to control the rate of their release, thereby offering a promising application for the development of sustained-release formulations. It has also been observed that to improve mucoadhesion to mucous surfaces, the building blocks of mucoadhesive systems are thiolated for covalent binding to mucin subdomains. In addition, porphyrins having an affinity for cancer cells can also be involved in the targeted action of drug formulations in the creation of mucoadhesive drug delivery systems.

The state-of-the-art is that the creation of effective drug delivery systems can involve mucoadhesive polymers, which can form supramolecular compositions with macrocycles. If the macrocycle in such a mixture provides binding and release of drugs, then the additional presence of polymer can not only increase an amount of encapsulated drug, but stabilizes the system as a whole and ensures efficient drug delivery across various mucous membranes. In general, mixed systems based on macrocycles and polymers perform better than individual systems. 

Regarding the potential clinical relevance of mucoadhesive systems, scientists are finding ways to develop them using various approaches to improve the bioavailability of less or ineffective drugs by manipulating composition formulations. The polymer and supramolecular approaches must be jointly implemented in order to find new mucoadhesive polymers and macrocycles with additional attributes of biodegradability, biocompatibility, non-toxicity, mucoadhesiveness for certain cells or mucous membranes for the successful delivery of molecules needed by the body. Although the physicochemical and biological properties of pure mucoadhesive polymers and macrocycles and the published results of clinical use allow them to be considered as promising raw materials for the preparation of drugs with various pharmacotherapeutic actions, the ability of formulations based on them to actually respond to medical needs has still to be demonstrated, and this imposes collaboration with medical researchers for potential clinical translations.

## 4. Conclusions

To conclude, this review addresses selected topics in the field of biomimetic amphiphilic systems including both fundamental and practical aspects, with studies over the last few years treated as a matter of first priority. The design and synthesis of amphiphilic building blocks is documented to attract wide attention, which is no little measure due to their practical usefulness. Since a wide variety of practical applications is aimed at the development of drug carriers, answering the green chemistry criteria is strongly required. Therefore, much effort was devoted to the preparation of low-toxic, biocompatible, and degradable amphiphiles capable of self-assembling in nanoscale aggregates at low concentrations. Special interest in cationic surfactants is due to their high affinity to biopolyanions and negatively charged membranes. Biomimetic principles in the design of novel amphiphilic compounds based on natural fragment (amino acid, sugar, bile salt, nucleobase) and environmentally friendly silicone moiety approves oneself as a promising strategy for construction of soft supramolecular systems with desired shape and stimuli-responsive functions. Specifically, this strategy has a great possibility for development of drug delivery systems with various morphology, tunable physicochemical properties and stability, which could become a useful tool for the solution of the problem of controlled release of cargo.

Liposomal formulations have received the main attention due to the fact that they were among the few carriers approved for the clinical treatment. Since numerous marketed liposomal therapeutics are available, these formulations appeared to be of particular importance for the analysis of bench-to-bedside factors. Several generations of liposomes were designed and actively advanced towards the market, beginning with conventional liposomes to stealth and targeted modifications, with the strategies for enhanced loading capacity, triggered release, endosomal escape, solution of PEG dilemma, and overcoming the BBB developed. Although specificity occurs in the development of different family of carriers, there are many common challenges typical for each of them. This is exemplified herein by such problem as endosomal escape that was discussed in detail, including ways for overcoming it. As alternative nanocarriers solid lipid nanoparticles, niosomes and cerasomes are identified that are assumed to provide the better standard of toxicity, biocompatibility, loading capacity and morphological stability in the near future. 

Analysis of the reasons preventing fast progress in the translation of the formulated drugs to the clinic reveals that they are mostly related to the manufacturing process and consist of the loss of product stability and quality upon scaling up of the preparation procedures, as well as in problems with the reliability and reproducibility of the final drug. To some extent, these drawbacks can be reduced by simplification of the preparation of the nanocarriers, excluding complicated synthetic procedures. Therefore, much attention is paid to the publications focusing on the noncovalent modification of nanocarriers, with the priority given to cationic surfactants, that revealed their beneficial efficacy from the viewpoint of colloidal stability, overcoming the biological barriers, cellular uptake, mitochondria-targeted drug delivery, etc. In this context, modification of nanocarriers with cationic additives is a promising strategy, especially with the use of charge-reversal amphiphiles that allow for preventing of lysosome degradation of encapsulated cargo. The disadvantage of cationic carriers is their toxicity that can be solved by tuning the lipid composition, with the balance between toxic characteristics and beneficial properties (stability, electrostatic affinity to biomembranes, antimicrobial effect) achieved. 

As a whole, the key point that should be stressed is the tendency to engineering the polycomponent formulations providing a synergetic effect. Hence, the colloidal, polymer and supramolecular approaches must be jointly implemented in order to find new effective nanocarriers for the successful delivery of molecules needed by the body. From this viewpoint, macrocycle–polymer compositions are of importance, since macrocycles exhibit selective binding of the drug resulting in enhanced solubility, while polymer shell guarantees the stability in biomicroenvironment and improved affinity to biointerfaces. Since the physicochemical and biological properties of some pure surfactants, mucoadhesive polymers and macrocycles, as well as the published results of their clinical use, present them as favorable raw materials for the preparation of medicinal compositions, the mixed compositions based on them can be considered promising for the design of effective drug nanocarriers.

## Figures and Tables

**Figure 1 ijms-21-06961-f001:**
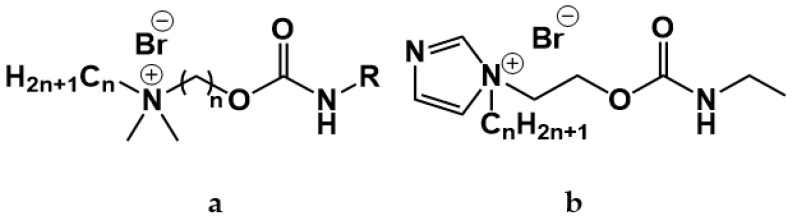
Chemical formulas of carbamate-containing amphiphiles bearing ammonium (**a**) and imidazolium (**b**) moieties.

**Figure 2 ijms-21-06961-f002:**
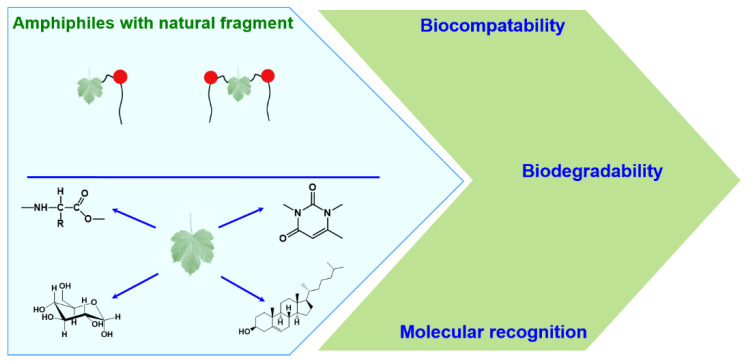
Schematic illustration of selected amphiphilic compounds bearing natural fragments and their benefits.

**Figure 3 ijms-21-06961-f003:**
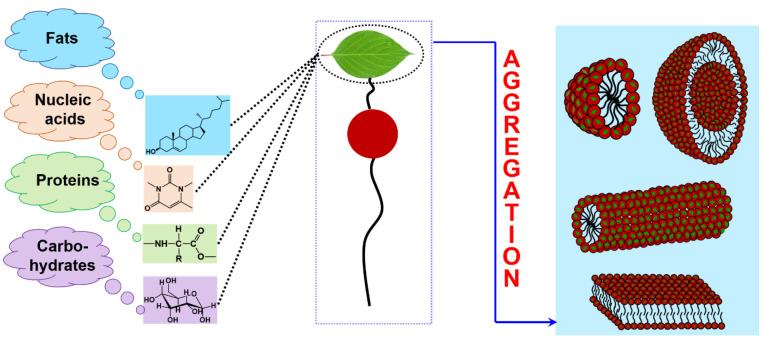
Schematic representation of nature-inspired amphiphiles and their possible various morphological structures.

**Figure 4 ijms-21-06961-f004:**
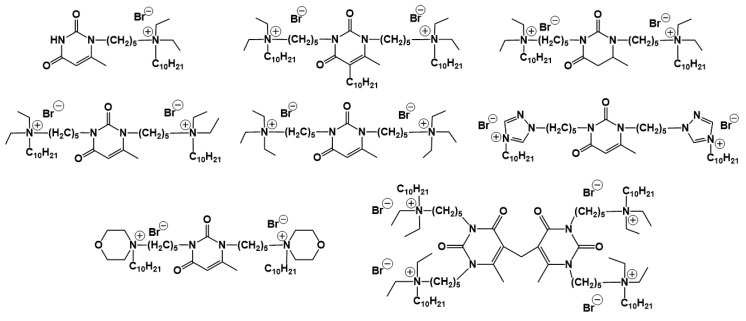
Some representatives of cationic pyrimidine-based nucleolipids [102,103,104,105].

**Figure 5 ijms-21-06961-f005:**
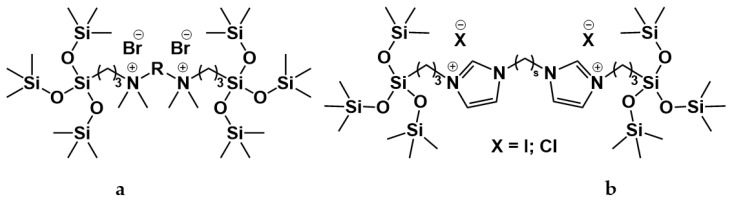
Chemical formulas of cationic tetrasiloxane gemini surfactants bearing ammonium (**a**) and imidazolium (**b**) groups.

**Figure 6 ijms-21-06961-f006:**
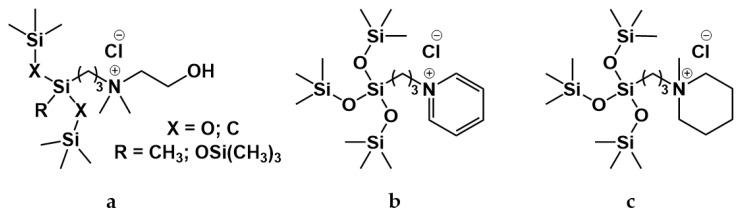
Chemical structures of cationic ammonium- (**a**), pyridine- (**b**) and piperidine-based (**c**) silicone surfactants.

**Figure 7 ijms-21-06961-f007:**
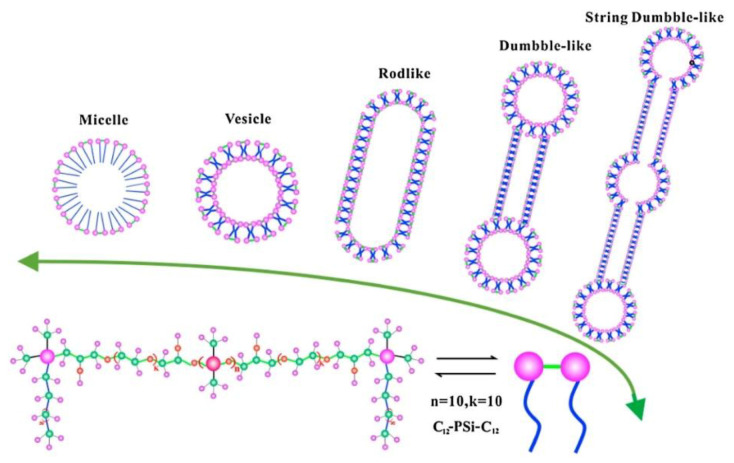
Possible models of morphological transition between aggregates of silicone-based surfactants at different concentration. Reprinted with permission from [123]. Copyright 2017 Elsevier.

**Figure 8 ijms-21-06961-f008:**
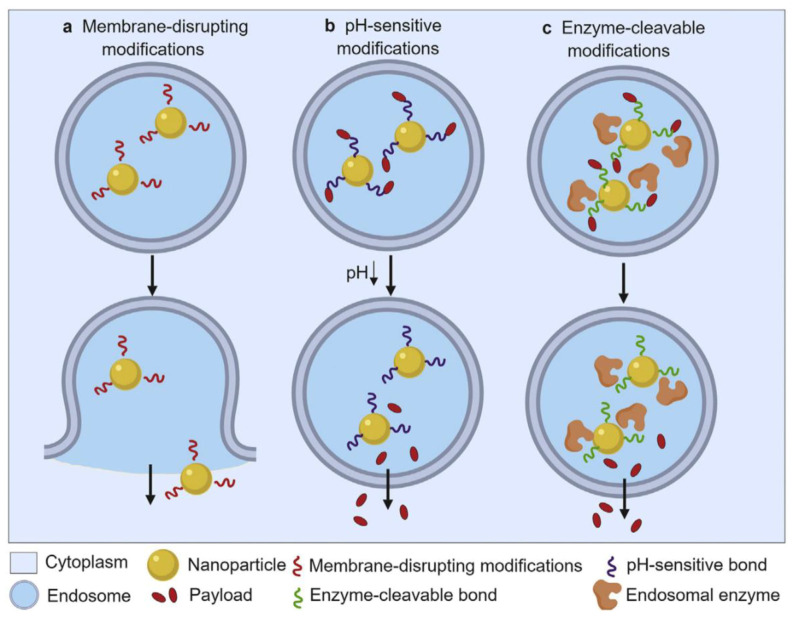
Examples of cytoplasmic delivery via endosomal escape. Three main strategies are available for nanoparticles to break through and escape endosomal barriers. (**a**) Membrane-disrupting surface modifications and mechanisms (e.g., poly(ethyleneimine) PEI; cell-penetrating peptides (CPPs); and lipid fusion with endosomal membrane); (**b**) pH-responsive materials (e.g., hydrazone bonds); and (**c**) enzyme-cleavable materials (e.g., ester linkages, cathepsin B cleavable peptides). Reprinted with permission from [174]. Copyright 2019 Elsevier.

**Figure 9 ijms-21-06961-f009:**
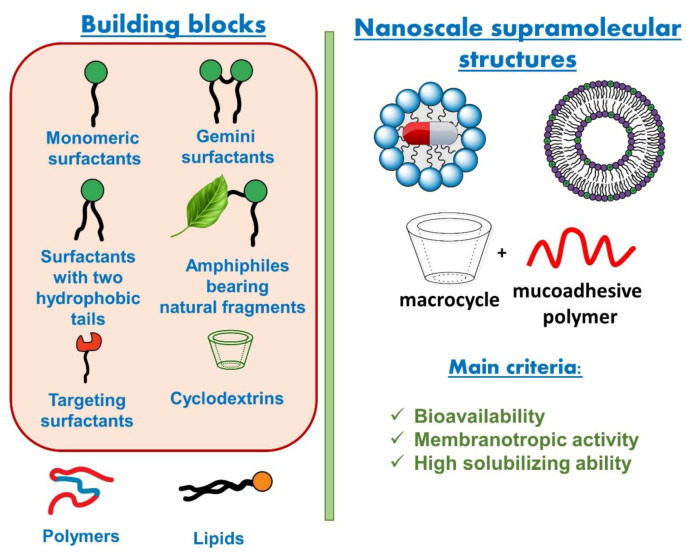
Schematic illustration of the building blocks and diversity of obtainable nanocarriers using non-covalent bilayer modification.

**Figure 10 ijms-21-06961-f010:**
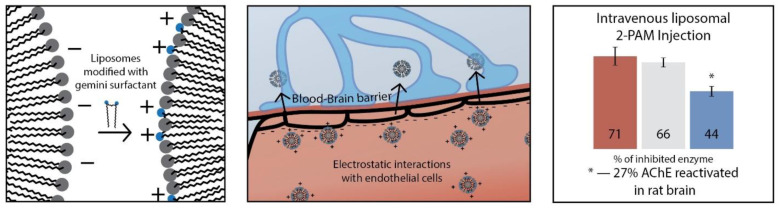
Electrostatic adsorption mediated transcytosis can be a mechanism to penetrate the BBB to reactivate brain AChE. Reprinted with permission from [216]. Copyright 2020 Elsevier.

**Figure 11 ijms-21-06961-f011:**
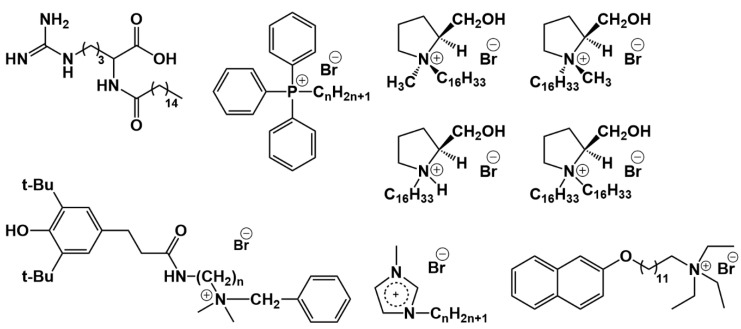
Structural formulas of single-tailed cationic surfactants.

**Figure 12 ijms-21-06961-f012:**
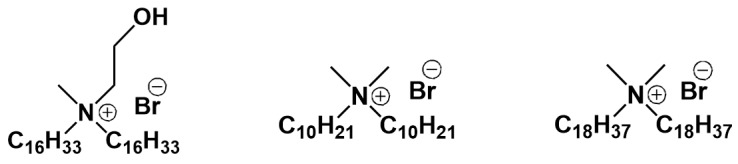
Structures of cationic surfactants with double, long chains.

**Figure 13 ijms-21-06961-f013:**
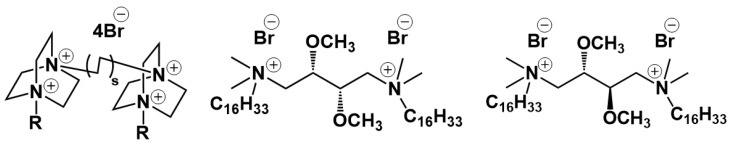
Structural formulas of gemini surfactants.

**Figure 14 ijms-21-06961-f014:**
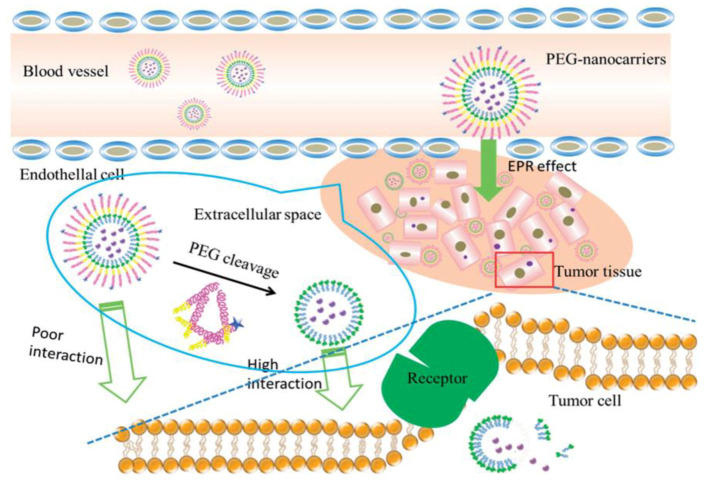
Principal mechanics of pH-sensitive PEG-shedding carriers. In the bloodstream, they utilize the stealth effect, then they accumulate in the tumor region due to the EPR effect, where a mildly acidic pH is responsible for PEG cleavage and enhanced absorption of the unpegylated liposome.

**Figure 15 ijms-21-06961-f015:**
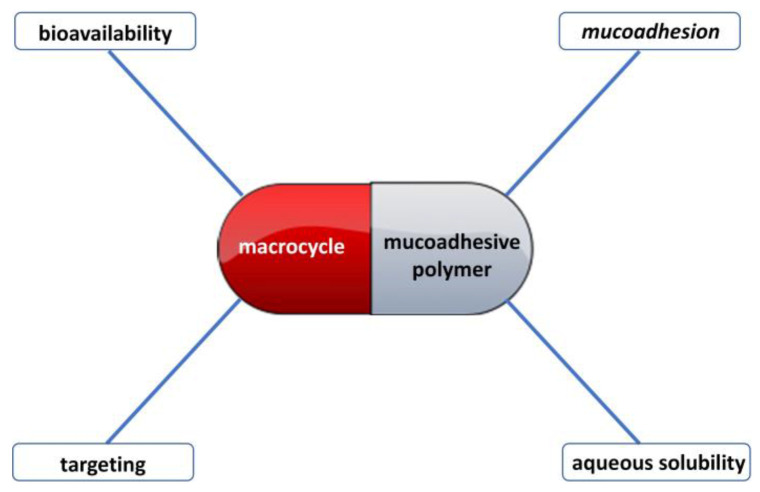
Graphic outline of the mixed macrocycle–polymer systems as mucoadhesive formulations.

**Figure 16 ijms-21-06961-f016:**
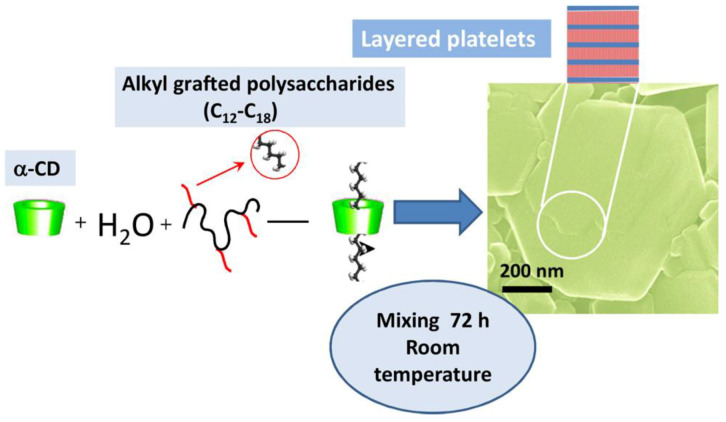
Schematic view of the complex formation between α-CD and alkylated polysaccharides followed by platelet organization in water. Reprinted with permission from [268]. Copyright 2018 American Chemical Society.

**Table 1 ijms-21-06961-t001:** CMC values and minimal inhibitory concentrations (MIC, μM) against *Staphylococcus epidermidis* strains for several amino acid-based surfactants.

Chemical Structure	CMC, mM	MIC, μM
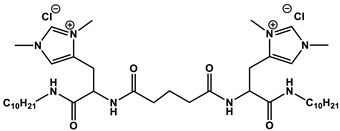	0.7 [47]	5.0 [47]
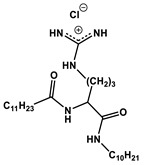	0.3 [40]	17 [40]
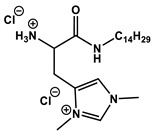	1.5 [45]	35 [45]
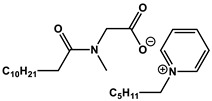	1.0 [33]	450 [33]
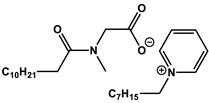	1.0 [33]	450 [33]

**Table 2 ijms-21-06961-t002:** Several examples of sugar-based amphiphiles with the lowest CMC values.

Chemical Structure	CMC, μM
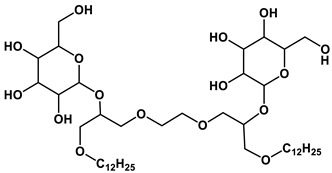	29.0 [80]
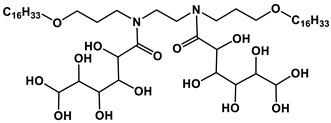	33.0 [76]
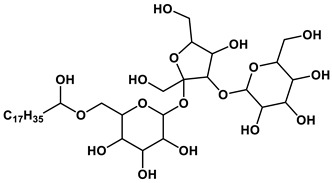	8.0 [69]
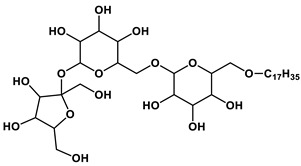	9.0 [69]

**Table 3 ijms-21-06961-t003:** CMC values and solubilization power *S* toward hydrophobic Orange OT for nucleolipids of various structures investigated in our reports.

Chemical Structure	CMC, mM	10^3^·S, mol_dye_/mol_amphiphille_
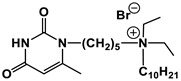	15.0 [106]	-
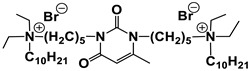	3.0 [107]	-
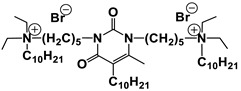	0.05 [108]	-
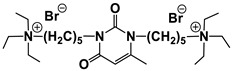	>100 [106]	
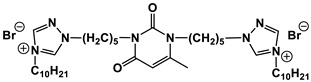	3.0 [102]	1.6 [102]
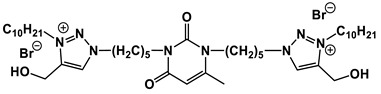	1.9 [103]	9.4 [103]
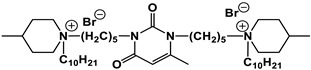	2.0 [109]	21 [109]
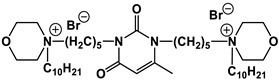	3.4 [104]	1.4 [104]
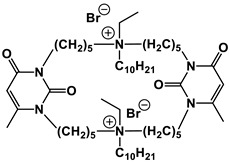	0.9 [110]	-
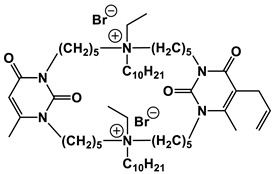	1.0 [111]	7.8 [111]
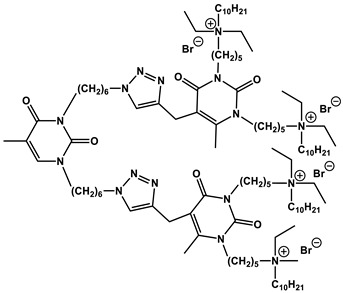	0.4 [112]	12.9 [112]

**Table 4 ijms-21-06961-t004:** Examples of novel successful approaches that utilize nanocarriers to solve typical chemotherapeutic and drug delivery problems.

Formulation Description	Problem to Be Solved	Solution	Ref
DOX liposomes	fast clearance by the reticuloendothelial system	Modification of liposomes with PEG-conjugated lipids, which hinder their recognition by macrophages. This provides prolonged circulation time of pegylated liposomes.	[142]
DOX liposomes	passively targeted versus ligand-targeted liposomes testing	Liposomes modified by folate demonstrated enhanced biodistribution in folate-expressing tumors.	[143]
Paclitaxel liposomes	improvement of loading capacity of hydrophobic drug	Incorporation of triglyceride increased the fluidity and lamellarity of the liposomes thereby resulting in sharp increase in concentration of drug loaded.	[144]
DOX liposomes	targeted delivery and cellular uptake	Protonation/deprotonation equilibria to switch the peptide between an “anchored” inactive position and active targeting position within tumor medium.	[145]
Irinotecan liposomes	PEG dilemma	PEG-shedding in lower pH achieved by attaching the PEG chain to the lipids via imide bond.	[146]
DOX liposomes	endosomal escape	DC-Cholesterol protonation in endosome media adds positive charge to the membrane and facilitates liposome-endosome fusion.	[147]
Porphyrin fumed silica–liposome nanocomposite	triggered release	Protonation of the fumed silica surface releases liposomal cargo.	[148]
DOX cerasomes	triggered release	Cerasomes were prepared with addition of thermosensitive DPPC and DMPC lipids.	[149]
DOX liposome	endosomal escape	Liposomes based on POPC and malachite green derivative carrying a long alkyl chain exhibited fusogenicity following UV irradiation	[150]
DOX polymer nanoparticles	endosomal escape	Proton sponge effect by binding protons present in the endo-lysosomes on the tertiary nitrogen atoms in the N,N-dimethylaminoethyl methacrylate units.	[151]
Paclitaxel mixed micelles	endosomal escape	Poly(β-amino ester)-mediated endosomal escape through proton-sponge effect.	[152]
DOX niosomes	targeted delivery	The glucosamine anchored DOX- loaded targeted niosomes showed the longer circulation in plasma with significantly higher bioavailability	[153]
Olanzapine solid lipid nanoparticles	overcoming the BBB	The formulated nanoparticles with olanzapine showed a significant increase in relative bioavailability, i.e., 23-fold in the brain compared to pure olanzapine suspension.	[154]
DNA liposome	endosomal escape	Cationic lipids destabilize negatively charged endosomal membranes through ion-pairing mechanism, causing a phase inversion.	[155,156]
Polymer nanoparticles	endosomal escape	For polymers (e.g., PEI, chitosan, PAMAM dendrimer) bearing ionogenic groups capable of being protonated at acidic pH additional mechanism is assumed referred to as proton-sponge effect.	[157,158]
Amphotericin B chitosan nanoparticles	oral and targeted delivery	An orally active nanomedicine based on an amphiphilic polymer nanoparticle with mucoadhesive properties provides a relative Amphotericin B oral bioavailability of 24.7%.	[159]
Telmisartan nanoparticles	instability due to aggregation, poor permeation through cornea	Addition of hydroxypropyl methylcellulose to γ-CD–drug complex	[160]
Celecoxib nanoparticles	low drug solubility	Addition of hydroxypropyl methylcellulose to CD–drug complexes	[161]
Celecoxib nano- and microparticles	low drug solubility, poor mucoadhesion and cytocompatibility	Addition of hyaluronic acid to randomly methylated β-CD–drug complex	[162]
Piroxicam tablets	weak drug release, poor permeation through buccal epithelium	Complexation with CDs was used to provide controlled drug release in vitro, and the additional combination with chitosan increased the permeation of the drug across buccal mucosa.	[163]
Simvastatin nanoparticles	low drug solubility, pure absorption and permeation through intestinal mucosa, fast drug release	The drug-loaded nanoparticles suspensions were prepared by ionotropic gelation method using chitosan, sodium tripolyphosphate, β-CD and coated with Eudragit L100.	[164]
Eugenol nano- and microparticles	fast drug release	Drug release from electrostatic CD–chitosan aggregates was prolonged due to aggregation ability in contrast to CD–chitosan conjugate.	[165]
Itraconazole micelles	low drug solubility, corneal tissue irritation from pluronics	Modification of the pluronics micelles through the incorporation of β-CD and polyethylene oxide.	[166]
Econazole nano- and microparticles	instability due to aggregation, low bioavailability	Combination of hydroxypropyl-β-CD with Tween 80 protected the encapsulated drug from aggregation. The suspension in chitosan acidic solution increased the drug bioavailability.	[167]
Clotrimazole nanofibers	fast drug release, poor mucoadhesion	Drug-loaded polyvinylpyrrolidone/hydroxypropyl-β-CD fiber was coated with chitosan-cysteine/polyvinyl alcohol.	[168]
Camptothecin nanocapsules	low drug loading, fast drug release, intestinal permeability	Addition of chitosan glutamate (PROTASAN™ UP G 113) to amphiphilic CD–drug complex	[169]
Timolol maleate compositions	pure permeation through the bovine cornea	Addition of hydroxypropyl-β-CD to bioadhesive polymers	[170]
DOX mesoporous silica nanoparticles	poor mucoadhesion	Functionalization of nanoparticles by thiol groups	[171]
Docetaxel nanoparticles	low drug loading, fast drug release, intestinal permeability	The anionic emulsion polymerization of isobutylcyanoacrylate was carried out in a solution of methyl-β-CD/drug inclusion complex.	[172]

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
