# Peer review of "Self-Assembly of Amphiphilic Compounds as a Versatile Tool for Construction of Nanoscale Drug Carriers"

_ijms, 2020, doi:10.3390/ijms21186961_

Round 1

Reviewer 1 Report

Dear author, I have read the review entitled, 'Self-Assembly of Amphiphilic Compounds as a Versatile Tool for Construction of Nanoscale Drug Carriers' with high interest. The article is well thought out and the important aspects of the topics are covered in details. Hence I am recommending to accept this paper after minor revisions. 1. While much importance has been given to demonstrate several amphiphilic compounds for potential nanoparticle carriers, authors did not show any figures related to the biomedical applications using these carriers. It will be great if you can include some published high quality figures here. 2. Please make a section to mention the challenges of these nanocarriers for potential clinical translations and how authors think to overcome those. 3. A table including the top examples from various sections will be a great addition to the manuscript. 4. Conclusion is very general and needs more authors input to better understand the future directions.

Author Response

1. While much importance has been given to demonstrate several amphiphilic compounds for potential nanoparticle carriers, authors did not show any figures related to the biomedical applications using these carriers. It will be great if you can include some published high quality figures here.

Response: Figures related to the biomedical applications using nanoscale drug carriers based on amphiphilic compounds have been added and presented by numbers 3, 8, 10 and 14. Figures reflecting the aggregation behavior in systems based on surfactants and macrocycle-polymer compositions as potential nanoparticle carriers have also been presented in the revised manuscript by numbers 7 and 16, respectively.

2. Please make a section to mention the challenges of these nanocarriers for potential clinical translations and how authors think to overcome those.

Response: Regarding the potential clinical relevance, a section 2.2.1 on this issue has been added in the revised manuscript. This section focuses on the clinical potential of liposomal formulations as there are examples of market entry for this type of drug carrier. Several generations of liposomes were designed and actively advanced towards the market, beginning with conventional liposomes to stealth and targeted modifications, with the strategies for enhanced loading capacity, triggered release, endosomal escape, PEG dilemma, and overcoming the BBB developed. Although the specificity occurs in the development of different family of carriers, there are many common challenges typical for each of them. Analysis of the reasons preventing fast progress in the translation of the formulated drugs to the clinic revealed that they are mostly conjugated with manufacturing process, and consist in the loss of stability and quality of product upon scaling of the preparation procedure, as well as in problems with reliability and reproducibility of final drug.

In terms of other drug delivery systems, scientists are finding ways to develop them using various approaches to improve the bioavailability of less or ineffective drugs by manipulating composition formulations. The colloidal, polymer and supramolecular approaches must be jointly implemented in order to find new effective nanocarriers with additional attributes of biodegradability, biocompatibility, non-toxicity, mucoadhesiveness for certain cells or biobarriers for the successful delivery of molecules needed by the body. Although the physicochemical and biological properties of some pure surfactants, mucoadhesive polymers and macrocycles and the published results of clinical use allow them to be considered as promising raw materials for the preparation of drug composition with various pharmacotherapeutic actions, the ability of formulations based on them to actually respond to medical needs has still to be demonstrated, and this imposes collaboration with medical researchers for potential clinical translations. This point has been added at the end of the revised manuscript.

3. A table including the top examples from various sections will be a great addition to the manuscript.

Response: Tables 1-3 on CMC and MIC values of surfactants have been added in the revised manuscript. Table 4, including the top examples from various sections, have also been added.

4. Conclusion is very general and needs more authors input to better understand the future directions.

Response: The conclusions have been rewritten to become gained in clarity and focus.

Reviewer 2 Report

This paper is an up-to-date review on the use of self assembling amphiphilic systems and their use as efficient carriers. Since two decades the use of biocompatible molecular systems capable of interacting together, through long-range (electrostatic) and short range basically hydrophobic) forces, to form aggregated nanoscale systems, has received much attention in chemistry, pharmacoly and medicine. The manuscript gives an overview of the amphiphilic systems elaborated by different adequate chemical fragments. A special attention has been focused on cationic carriers, by mentioning their phsico-chemical properties, as well as their adavantages and disadvantages (toxicity).

As a conclusion, this paper may be of interest to a large audience composed of phsico-chemists, chemists and pharmacologists, and its publication is strongly recommended.

Author Response

Thank you very much for reviewing our manuscript and for providing supportive commThank you very much for reviewing our manuscript and for providing supportive comments.ents.